# Climbing *the Giara*: A quantitative reassessment of movement and visibility in the Nuragic landscape of the Gesturi plateau (South-Central Sardinia, Italy)

**Davide Schirru, Alessandro Vanzetti** [ORCID] *

Dipartimento di Scienze dell'Antichità, Università degli Studi di Roma "La Sapienza", Roma, Italy

* alessandro.vanzetti@uniroma1.it

**Data Availability Statement:** All relevant data are within the paper and its Supporting Information files.

## Abstract

The built landscape of Nuragic Sardinia is an exceptional case for geostatistical analysis, allowing for a discussion of long-held assumptions and ideas. The function of nuraghi (ca. 1700–1100 BCE), the most prominent settled monument of the Sardinian Bronze Age, has been addressed via a multiplicity of landscape approaches, mainly relying on intuitive assessments of their spatial properties: nuraghi were assumed as means of territorial control. The series of nuraghi crowning the mesa plateau named *Giara* of Gesturi (South-Central Sardinia) provides a significant case for the study of their relations with visibility and movement. Context-oriented GIS models based on viewsheds and least-cost paths have been devised as targeted tools. The results show a certain correlation between nuraghi and potential movement on the slopes, thanks to the selection of plateau morphologies such as outward crests. Anyway, nuraghi do not stand exactly at the most accessible points of the plateau. Nuraghi offered ample visual control, especially at large distances, but not specifically over the closest accessible ways. This suggests that the function of nuraghi is somehow connected to defense and visibility, but it is not explained directly by local territorial control: a role as landmarks and multifaceted monuments has likely to be envisioned.

## Introduction

### Sardinia as a geospatial laboratory

The potential of the archaeology of Sardinia in shedding light over the processes of Mediterranean deep history has been increasingly remarked by recent research [1]. Sardinia is one of the major islands in the Mediterranean sea, in a rather central position; it is remote and close enough to interposed islands and to the coasts of mainland Europe and Africa, to have been selectively approached through millennia, and is characterized by peculiar resources, such as obsidian and metal ores. These characteristics highlight its potential as a laboratory both for island colonization and connectivity, and for internal -rather secluded- cultural transformation processes [2]. This double feature, at the same time of isolation and connectivity, has been

**Funding:** The authors received no specific funding for this work.

**Competing interests:** The authors have declared that no competing interests exist.

further enhanced in general anthropological perception, thanks to the contribution of aDNA to the study of human biological descent and mobility: Sardinia has emerged as one of the most conservative places in Europe and the Mediterranean, in terms of genetic continuity from Early Neolithic times to the end of the Nuragic Bronze Age, thence supporting the idea of a macro-context of basically internal developments. Albeit further research will be necessary to substantiate in detail these considerations, which have been achieved in the frame of the so-called "third science revolution" of archaeology [3], they result from two different and independent studies on samples of similar size, apparently reflecting a somehow robust pattern [4, 5]. Indeed, it is likely that this image of Sardinia as a separated enclave is somehow deceptive, in front of the specific marks of contacts and connectivity, visible at least in the coastal fringe: interaction processes seem to have been intense and significant, but mainly acting as localized primers, and stimulating prevailingly local internal transformations.

In this sense, we guess Sardinia, particularly during Nuragic times of the Bronze and Early Iron Ages (ca. 1.700–700 BCE [6–8] can be assumed as a well-suited case for the study of human choices and developments, in the frame of a structured society and rather visible archaeological evidence. The possibility to test models on empirical data recorded in the landscape is beyond most other archaeological situations in the central Mediterranean, notwithstanding some persistent uncertainties in functional and chronological detail of the archaeological sites that will be discussed afterwards. Comprehensively, the distribution of Nuragic monuments in the recorded landscapes provides a palimpsestic view of cumulative choices made by ancient communities that allow to develop proper quantitative and geostatistical approaches.

We will first review past and current theoretical and methodological approaches to Nuragic landscapes, in order to reconstruct how interpretations of the functional role of nuraghi have come about (below, § From monuments to Nuragic landscapes). A specific paragraph will be devoted to the discussion of the different observed uses of GIS in the study of Nuragic landscapes (below, § Theoretical issues and landscape approaches).

In this study we select the area around *the Giara* di Gesturi (hereafter quoted as: *the Giara*) a wide mesa-type basaltic plateau located in South-Central Sardinia, with a characteristic distribution of nuraghi and villages of the Bronze and Iron Ages (below, § Materials).

A series of quantitative GIS-based models, aimed at the assessment and quantification of commonly accepted ideas regarding the spatial properties of nuraghi, will be devised and employed in order to explore context-specific patterns of movement and visibility in *the Giara* and its surroundings (below, §§ Methods; GIS Analyses).

Specifically, this will entail:

- the building of a generalized movement model, based on the generation of least-cost paths with randomized origin, directed towards *the Giara* plateau (mobility/accessibility);

- the use of viewsheds, centered on the existing nuraghi, as a means to investigate the degree of visual control over the reconstructed movement pattern (visibility).

The results will allow us to compare the mobility/accessibility vs. visibility properties associated with *the Giara* nuraghi with those associated with points randomly generated in analogous positions, in order to test for their significance.

By arguing that these quantitative, hypothesis-testing models provide an effective way of addressing current intuitive ideas developed for Nuragic landscapes, we will provide an assessment of how the function of nuraghi in the specific context of *the Giara* can be better conceptualized. These results can also provide a conceptual framework to tackle spatial trends across the island in a more sound way than the still dominant intuitive and descriptive interpretive

proposals of settlement patterns, even when using GIS as a tool. We hope in this way to contribute to strengthen the use and evaluation of formal modelling (cf. [9]).

## From monuments to Nuragic landscapes

### The archaeology of Nuragic Sardinia

The definition and name of the Nuragic culture stem from its most recognizable monuments–the nuraghi–which still constitute an integral and highly visible part of the contemporary landscape of the island [8, 10–12]. The archaeological features associated with the Nuragic culture, developing at least since the start of Sardinian Middle Bronze Age (ca. 1.750 BCE) have been recognized across the entire island, as a testimony of its widespread diffusion. They are a variety of well-defined monumental types as well as numerous shared traits of material culture [13, 14]. Monumental buildings are represented since the start mainly by the nuraghi towers in their various forms, along with the collective burial monuments known as Giants' tombs [15–20]. Dry-stone hut villages represent another significant component of Nuragic landscapes, present since the start but growing in number and relevance particularly during the later stages of the Sardinian Bronze Age (after 1.100 BCE) and continuing in their use in the Early Iron Age (950 BCE onwards). Probably at least since the Recent Bronze Age (ca. 1.350 BCE), markedly ritual monuments such as "sacred" wells and springs dot the landscape [13, 21]. The resulting landscapes are populated by structurally and likely functionally different sites, rather easily classified in distinct categories, albeit showing significant case-specific variability. The human effort underlying the construction of Nuragic buildings and landscapes is a testimony of structured social and economic dynamics, whose powerful rise, during the middle phase of the Bronze Age (ca. 1.750–1.350 BCE), is counter-matched by the deep crisis and transformation of Nuragic culture in its later stages, in the final phase of the Bronze Age (ca. 1.200–950 BCE) and during the Early Iron Age (950 BCE onwards) [22]). Indeed, the Nuragic phenomenon can be considered as one of the most recognizable–in terms of scale and distinctiveness–in Mediterranean prehistory, leading to a sustained interest from Italian and international scholars (e.g. [1, 2, 8, 10, 12–14, 22–33]). The specificities of Nuragic cultural and social aspects seem to reflect a peculiar development inside typically insular phenomena of segregated development, leading to the selection of singular traits. The internal affinities in Bronze Age Sardinian cultural manifestations and monumental landscapes are much stronger than we can remark in Sicily, the only Mediterranean island with a comparable size [34, 35], and possibly also than close-by Corsica, where the growing comprehension of the monumental record of tower-like buildings shows also their considerable variability, as well as their distribution only in the southwestern part of the island [36, 37]. The visibility of its archaeological features further makes Sardinia an emblematic case of Mediterranean prehistoric developments, worth checking in its dynamics, as it challenges simple interpretations [38]. The discussion about the social milieu of Bronze Age Sardinia makes generally use both of the funerary rituals characterized by the monumental collective Giants' tombs [15,17–20,39], gathering dozens of mainly adult individuals with, in most cases, not particularly impressive grave goods [40, 41] (with, however, some exceptions, as recently shown [16]) and of the outstanding landscape packed with towering buildings [8]. With regards to the social interpretations of MBA and RBA Nuragic communities, a number of ideas have been put forward. A more traditional view, established with the work of Lilliu [14, 42] but still embraced, and even accentuated, by some scholars (e.g. [43, 44]) sees Nuragic communities as ruled by warriors/chiefs: in these reconstructions, nuraghi acted both as fortresses and chief houses, in a way akin to medieval castles. In subsequent reconstructions, the complexity and hierarchical component of Nuragic communities has either been downplayed [30] or nuanced, with more emphasis put on the dialectic

between community linkages and emergent centers of power as testified by the differential architectural complexity of Nuragic buildings [26, 28, 45–47]. In fact, the construction of complex nuraghi has been envisioned as the outcome of ever-increasing competition dynamics in otherwise cooperative communities [48]. According to Araque-Gonzales, cooperative processes, rather than top-down, coercitive ones, can explain the construction of both complex nuraghi and later sanctuaries [49, 50]. In a general sense, these perspectives can be summarized as basically referring to the hierarchic/heterarchic opposition, but looking at details, they are considerably more complex.

## Nuraghi as buildings

Nuraghi are widely spread in the different areas of the island, with an estimated number ranging around 7000 units, or even more [8, 32, 51]; they have been at the center of most research and debate concerning the Nuragic civilization, as their role is undoubtedly a vital component of this phenomenon. The origins of the nuraghe are still disputed, and the relevance of previous monumental architecture for its development, such as the round building of Monte Baranta's "tower-hut", dating to the Eneolithic phase of Monte Claro (second half of 3rd millennium BCE), is still a matter of debate [6, 52]. In fact, limited settlement data is available for the Early Bronze Age [13, 53], which is in turn mainly represented by funerary sites.

The first appearance of monumental features commonly classified as of Nuragic type is dated, using $^{14}$C dates, alternatively to the Sardinian Middle Bronze Age 1 (MBA1 ca. 1750/1700-1600 BCE) or MBA 2 (ca. 1600–1500 BCE) [8, 13]. The earliest monuments are defined as protonuraghi or "archaic nuraghi" ([27, 54–56] for a discussion of the terms), i.e. stone-built monumental platforms showing a wide range of architectural solutions in terms of plan and internal subdivisions of the platform basement: they include corridors and truncated corbel-vaulted chambers. For this reason the buildings have sometimes been named corridor-nuraghi. More complex examples of archaic nuraghi are also known, arranged on multiple levels or storeys [31, 43, 57]. The first construction of stone-built, collective burials known as Giants' Tombs can also be dated to this first phase [26, 41]. Additionally, at this early stage very few hut villages, some of which built in perishable material, are known [58, 59]; typical are rectangular huts, sometimes massive in their walled structure.

It is starting from the MBA2 or the MBA3 (ca. 1500–1350 BCE) until the whole Recent Bronze Age (RBA: ca. 1350–1200 BCE) that the large majority of Nuragic buildings–among which most of nuraghi of the tholos-type–can be dated [11], even if some authors have suggested an earlier start [6, 13]. Typical tholos nuraghi are based on the tower building model, made of one or multiple superimposed, corbel-vaulted chambers. Simple nuraghi are constituted by a single tholos tower, while 'complex nuraghi', show the presence of two or multiple tholos towers variously arranged around a central one, which is generally the highest and greatest in size; using a medieval analogy, it is defined as "the keep". Simple and complex nuraghi both spread as contemporary, related building types. The widespread and dense diffusion of Nuragic sites during this period (ca. 1500–1200 BCE) is often explained as a planned effort directed towards the exploitation of the natural landscape, which is reflected in the proximity of nuraghi to natural resources and productive soils in particular [22], as well as directed towards the control of key areas in the landscape, such as natural passageways, rivers and crossings [60, 61]. The expansion of the Nuragic phenomenon in the RBA (1350–1200 BCE) is paralleled by the increasing diffusion of stone hut villages [62], which are often built in the surroundings of complex nuraghi, and by the spread of monumental water facilities, springs and wells with a mixed secular and cultic function [21]. The Final Bronze Age (ca. 1200–950 BCE) and Early Iron Age (ca. 950–700 BCE) are generally regarded as the latest stages of the full-

fledged Nuragic civilization [24, 63] ([6, 42] for a different view), and witness profound transformations at the settlement and landscape level. In particular, a process of nucleation corresponds to the abandonment of most of nuraghi. At the same time, nuraghi seem to lose much of their original function, they apparently are no more built, but only maintained and transformed; often they host a range of cult-related activities [64, 65]. Population seems to concentrate in fewer, larger villages; in parallel with this phenomenon, the sites interpreted as communal sanctuaries develop and flourish [22, 24, 66–69].

Various functional meanings have been attached to nuraghi, which definitely contain occupation layers of residential type, with findings attributable to everyday and subsistence activities [70–73]: a traditional view interpreted them as fortresses being part of defensive systems recognizable at landscape level [10, 31, 74]. In most recent accounts nuraghi are more generically thought to be part of landscape control strategies [6, 24, 28, 33, 75], not to be intended in a strictly "militaristic" sense: Webster 1996 straightly calls them "farmsteads". Beyond merely functional interpretations, symbolic aspects linked to their construction have also been emphasized [30, 45, 76, 77]. In particular, the symbolic interpretations of nuraghi vary according to the overall social reconstruction: for example, they have been thought to symbolize the power and wealth of single families [30] or, at the other end of the spectrum, the cohesion of entire communities, which collectively contributed to their construction [77]. These interpretations have drawn upon architectural analysis, excavation data and–centrally to the theme of this paper–the spatial arrangement of nuraghi across the landscape.

## Theoretical issues and landscape approaches

### Theories and methods in traditional Nuragic landscape studies

The development of landscape and spatial approaches in the study of the Nuragic civilization has been characterized by a distinct and unique trajectory, that has set it apart–for most of the twentieth century–from the variety of theoretical and methodological trends developed elsewhere in Europe, and still influences modern analyses. A brief introduction to the development of this trajectory is therefore necessary for a fuller understanding also of contemporary perspectives and challenges.

The work of Antonio Taramelli [78, 79] set the methodological end epistemic foundation of later studies, while also providing specific interpretations of Nuragic landscapes that are still relevant in current debates. At the core of his approach lay the belief that the functional properties of monuments, and specifically of nuraghi, could be inferred from their architecture, their relationships with the physical environment, as well as from their reciprocal distribution. With this aim in mind, the highly preserved and visible Nuragic landscapes provided ideal case-studies, and particularly so the *giara* plateaus of Gesturi [78] and Serri [79], where he employed a combination of monument-centered fieldwork and cartographical intuitive assessment. The distribution of nuraghi was interpreted as following and enhancing the *natural* features of these particular local morphologies, and namely their potential defensibility and visual control over the surrounding landscape. Such features were treated as self-explanatory traits, that would emerge upon investigation–thus attributing diverging interpretations to a mere lack of sistematic fieldwork. This approach opened the way for the long-held conception of nuraghi as fortified settlements, aimed at the military control of the landscape [74]: this idea was readily adopted by Lilliu [80] in his work on the smaller basaltic plateau of Siddi, as well as in his first synthesis on Nuragic towers [10]. In turn, this spatially-constructed idea constituted a major component of his militaristic view of Nuragic society as ruled by a class of "shepherd-warriors" [81]. Nuragic societies were also thought to be organized in territorial districts named cantons, each enjoying political independence [82]: as in Taramelli's approach, this

idea was based on an intuitive assessment of Nuragic settlement distribution–which appeared to be arranged in clusters–and their interpretation as political units resulted as a circular and self-evident correlate of their spatial arrangement.

The idea of Nuragic landscapes as reflecting wider social and functional dynamics has been echoed in recent syntheses [83] as well as in regionally-focused research, such as that conducted in the Sinis peninsula [84] and in the Montiferru massif [85], both located in Central-Western Sardinia. Here the militaristic view of nuraghi is rejected (see also [45]), and the monuments are considered as geared towards the agricultural exploitation of their surroundings, as part of a process of dramatic demographic expansion witnessed since the beginning of MBA3. In these otherwise fundamental contributions, the legacy of Taramelli's and Lilliu's approach is however clear in the treatment of spatial data: territorial systems are ~~intuitively~~ defined through an empirical cartography-based assessment of spatial clustering [84] or local geomorphology [85]. At the same time, the spatial relationship between nuraghi and agricultural soils, which certainly was an important factor in these landscapes [86], is only rarely systematically assessed. The general relevance of this spatial relationship has been highlighted by some more formal approaches [22] and a formalized assessment could therefore give more strength to the overall interpretation of the function and social meaning of these monuments.

Some of the most significant traditional approaches can be found in the discussion of the territorial system of Nuraghe Nolza in Central Sardinia [87, 88] and in the interpretation of the Nuragic landscape of the basaltic plateau of Pranu 'e Muru, centered around the major nuraghe Arrubiu [29, 60, 89]. Central to these discussions are the concepts of *visual dominance* and *landscape control*, which would be reflected in the locations chosen by Nuragic builders: such places seem to afford, for example, proximity to and visual control of access routes to plateaus, rivers and fords. A final example comes from the Cabu Abbas plain in North-Western Sardinia [75], with the analysis of the Nuragic settlement system centered around the complex Nuraghe Santu Antine: a key motif of this work is the assumption of locational properties of nuraghi in relation to "natural" pathways, soils and resources. The adoption of these approaches has generally been justified by the Authors by quoting the fluid and unique nature of each specific Nuragic landscape, which -they state- cannot be adequately described and explained with the use of rigid and abstract models, such as Thyssen polygons or Nearest Neighbor Analysis [61, 88, 90], and therefore neither with more complex geostatistical procedures. While we agree that the rigidness of certain models may not prove useful for addressing the complex archaeological questions posed by each specific Nuragic landscape, a model-based approach can and should also be flexible and context-oriented, as argued in the next section.

Finally, a recent approach addressing Nuragic landscapes is represented by the work of the Spanish school of Granada, with Sardinian associates [33, 91–94], based on the methodology devised by Nocete Calvo [95, 96]. These works rely on a model-based approach, whereby landscape properties such as visual control and defensibility are codified in a series of topographical indexes, such as related to topographical location (height and relative dominance in terms of differences in elevation), to geomorphological units (generally to be defined qualitatively, before indexing) and indexes related to different radii traced around the nuraghi, in a circular way (at 250 and 1000 m). These indexes are associated to monuments and then inspected through multivariate analyses. The structure derived from these analyses is used as a baseline for functional and social interpretations, mainly geared towards the assessment of social complexity as inferred from landscape hierarchization ([33], for a recent synthesis). The attribution of the scores is therefore crucial for the production of meaningful results, and here some problems may rise, as discussed for the case-study of Dorgali Municipality [94].

The suitability for different kinds of analysis of the incredibly visual landscapes of Nuragic Sardinia, has therefore been assumed since long time: the self-evidence of nuraghi makes them

a case-study of high and wide-ranging potential. Their relevance for the interpretation of Mediterranean societies, acting in the connected Late Bronze Age and characterized by long-distance overseas trading contacts [38] is promising: in fact, while the Nuragic social structure can be considered in its uniqueness, it can also provide–given the abundant monumental and spatial data–a comparative model for less known or preserved Mediterranean contexts. Moreover, a contextual analysis of spatial, architectural and formal spatial data, could provide key information for the reconstruction of social dynamics such as the emergence of hierarchical structures. For these reasons, we argue that some of the dominating intuitive or schematic assumptions that have generally ruled over the more or less formalized attempts of analysis have so far limited the significance and the comparability of the interpretation of Nuragic landscapes.

## Theory and GIS in the Nuragic contexts

The use of GIS as an archaeological tool can nowadays be considered commonplace, and its potential applications have already been summarized in numerous contributions [97–100]. Its initial adoption can be traced back to 1990's, in a period when formal spatial analysis stood at odds with the trending contemporary postprocessual theoretical frameworks [101]. In fact, the use of GIS seemed to represent a problematic return to the theoretical and methodological concerns of the New Archaeology, and was therefore considered at risk of environmental determinism ([102] for a discussion), given the type of data that would be normally incorporated in GIS analyses, as well as the "modernist" way in which these data are represented. Such critique stemmed from postprocessual, interpretive views of landscape, and phenomenology in particular [103–106]: from this point of view no GIS–and no formal analytical method in general–can be a substitute for a direct, embodied experience and understanding of the landscape, seen as an irreducible social and cultural construct. What has followed is a number of contributions trying to address these criticisms, by setting new theoretical frameworks, e.g. adopting the bottom-up perspective offered by agent-based modelling, where the concept of individual agency can be incorporated in full-fledged computer simulations (e.g. [107]). Other authors have also tried to accommodate postmodern views of space, place and landscape [108–112], while at the same time putting formal GIS-based models at the center of their methodological efforts. As it has already been noted [101], such contributions, while providing strong conceptual frameworks, have for a long time been ignored by non-GIS practitioners, leading to the development of new original theory (see also [113]). A further strand of contemporary spatial archaeology which adopts a more explicit model-based perspective, either through GIS or through other geostatistical tools (mainly R), can be viewed as a critical reappraisal of some of the New Archaeology concepts and methodologies, nowadays made viable thanks to significant hardware and software advances (e.g. [114–116]). The contemporary methodological and theoretical landscape remains therefore somewhat fragmented, even though contributions towards commonly accepted ways of knowledge production have been made [117].

The theoretical landscape in which the use of GIS and formal spatial analysis in Nuragic archaeology should be contextualized differs greatly from mainstream trends found in North American and British landscape archaeology. In first place, while some examples of a more formal hypothesis-testing approach can be found [118–121], the tenets of New Archaeology have never been significantly represented in the reconstruction of Nuragic landscapes, and the adoption of methodologies that have mainly been employed in that context often betray a much different scope [86, 122, 123], whereby these are used as descriptive tools rather than hypothesis-testing devices ([124, 125] for further examples). It is perhaps in the light of this observation that the criticism levelled by Sardinian scholars against formal spatial analysis can

be better understood: as tools of description, formal spatial models do inevitably fail to account for the idiosyncrasies shown by specific Nuragic landscapes, and set as a binding limit by researchers' interests (see also [85]).

At the same time, the set of analyses currently employed by the Spanish scholars (the Granada way) studying Nuragic Sardinia is in our view problematic as well: while providing explicit mathematical models–which undeniably favor their testability–these come with assumed cultural and functional meanings [94], which in turn become an essential part of these particular narratives. In other words, these approaches allow only for a comparison of spatial properties between different clusters or types of sites, leaving unanswered the question of the real significance of the spatial parameters adopted.

Despite the multitude of approaches characterizing most of Sardinian landscape archaeology, it can be argued that they share a common ground, given by the treatment of spatial data as informative of past cultural processes. It is clear that spatial data is in fact critical for any (re) interpretation of Nuragic culture, as showed by its integration in most reconstructions. At the same time, spatial data has in most cases been treated in an empirical way: ideas about the spatial organization of Nuragic communities–and of its social correlates—stem from and are formed as a result of observation, and contribute in this way to the formation of interpretive models, which are less often subjected to further testing. Such an approach has been applied in the discussion of the differential spatial properties and functions of simple and complex nuraghi [75, 88], in the reconstruction of the spatial extent of Nuragic territorial systems [60, 61, 85, 89, 90, 122] and of the relationships between nuraghi and environmental variables such as soils (e.g. [90]). In these contributions, despite the emphasis often put on context and its peculiarities [12], the approach is not informed by the postprocessual critique of landscape, whereby the emphasis is placed on the subject (the scholar) as a producer of knowledge and on the situatedness of any understanding (e.g. [104]): this reliance on the informative potential of empirical spatial data can and should allow for a fruitful dialogue between traditional and formal approaches, as opposed to a mutual rejection stemming from antithetic epistemological underpinnings.

However, while a commonly shared set of principles and trust in the sensible evidence of landscape does exist between Nuragic empiricist scholars and more formalized landscape researchers, leaving aside extreme postprocessual negativity, formal models are sometimes regarded with some suspicion, and described as too rigid to effectively tackle the issues raised by Nuragic landscapes [61, 89, 90]. Here we argue instead that the use of models as applied to the studies of Nuragic landscapes is not aimed at the simplification of otherwise complex spatial and functional dynamics but, rather, at establishing a shared frame of reference against which various hypotheses can be tested: among such hypotheses, a significant weight has to be attributed to those already established in the empirical research tradition, as they are suitable for testing.

A way forward can possibly be represented by what Llobera [112] defined as *scaffolding models* or *methods*. These are meant to be *ad hoc* models whose "aim is not to model some systemic property or produce a final insight about society. Instead, they are constructed to explore how particular processes or concepts may play out within the specifics of a certain context" [112]. Such kind of models seems to us particularly suited for the study of Nuragic landscapes, because they a) are flexible enough to account for the already noted variability among different Sardinian subregions, and therefore to address context-specific dynamics and b) may help reinvigorate current debates, as a form of middle-ground solution between current approaches. What is then needed is a process of *translation*, by which current intuitive assessments of spatial properties can be embedded into a GIS for analysis: in this process, intuitive models become quantifiable and, rather than providing a mere description of the context at

hand, they allow for the formulation of hypotheses that can be tested against data. This process is in no way straightforward or univocal, but its very nature allows for a multiple-step discussion of the spatial concepts at hand, as well as disentangling their complexity, often hidden in traditional intuitive approaches.

## Materials

### Study area: Nuragic landscape of *the Giara* of Gesturi (South-Central Sardinia)

In Sardinia, the term *giara* refers to mesa-type plateaus originated from plio-pleistocenic volcanic activity, found in the South-Central portion of the island [126]. Together with the Pranu 'e Muru and Pranu 'e Siddi plateaus, the *Giara* of Gesturi ([Fig 1]) represents the most notable example in the island, with an overall extension of 44sqkm and a long axis of 11km, with a NW-SE orientation. The Gesturi plateau, together with the other examples of Sardinian basaltic plateaus, has proved since the beginning of the 20<sup>th</sup> century as an ideal case study for addressing the cultural and functional meaning of nuraghi, as reflected in their spatial arrangement. Firstly, this fortune can be attributed to its nature of "closed context", whereby its

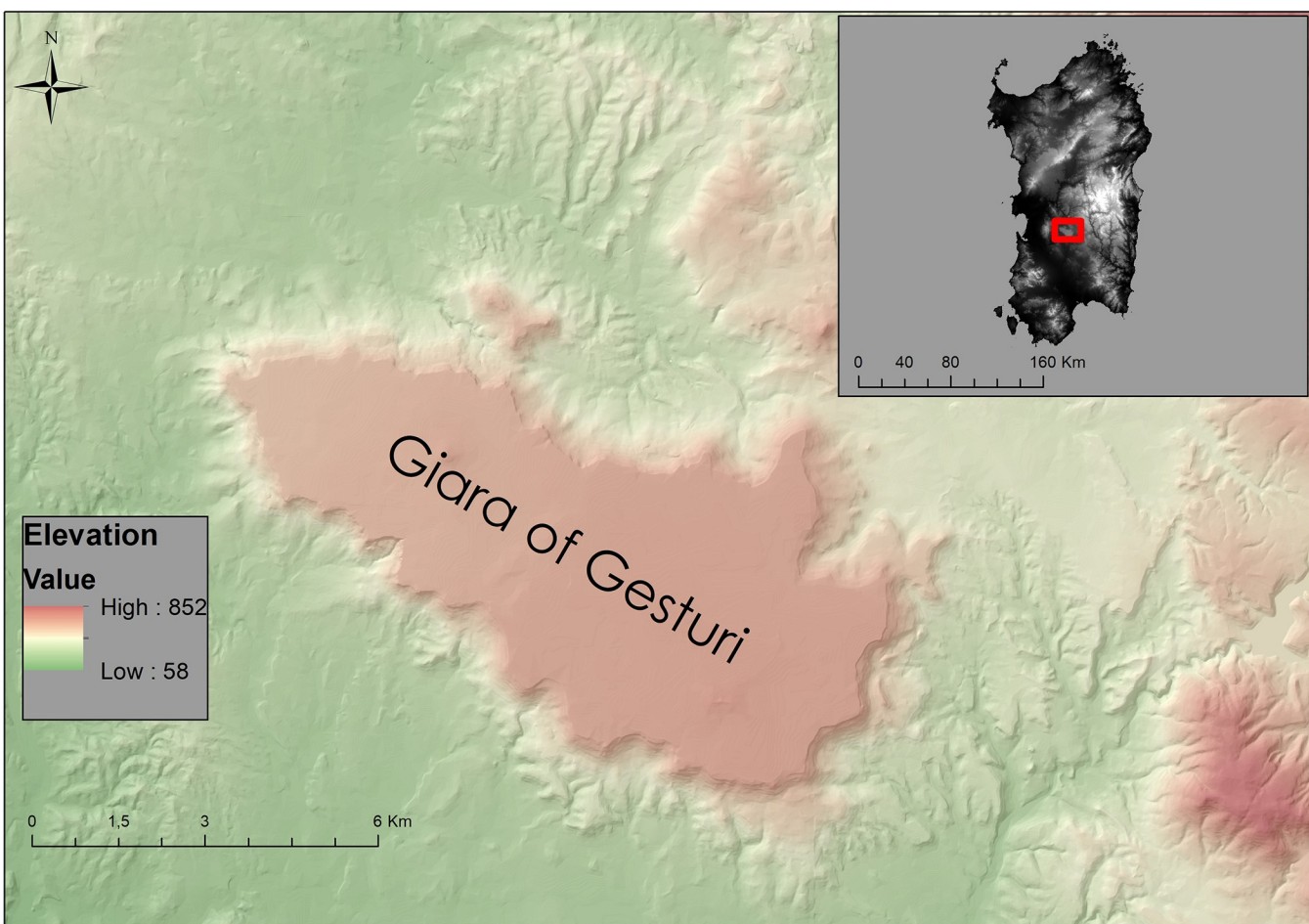

**Fig 1. Location of the mesa plateau of *the Giara* of Gesturi, South-Central Sardinia (Base map: Digital terrain model of the region of Sardinia, 10 m resolution; available at <https://webgis2.regione.sardegna.it/geonetwork/srv/ita/catalog.search#/metadata/R_SARDEG:JDCBN> under the CC BY 4.0 license; shapefile by the authors).**

morphology is well defined by abrupt slopes, making it an island-like section of the landscape. This specificity would have also been obvious to Nuragic communities: in fact, a notable feature of the nuraghi of *the Giara* is their peculiar spatial distribution. As already noted since the early work on the plateau [78], nuraghi are located exclusively on the very edge of *the Giara* and on its slopes, the flat inner mesa surface bare of towers. Other Sardinian plateaus share the same configuration, as is the case of the smaller examples of Siddi [80], Serri [79] and Guzzini [89]. In the Pranu 'e Muru plateau [60], nuraghi are instead found both on the edge and in its inner areas. While this variability could at least in part be caused by specific local preservation of nuraghi or by biases in the extensive survey techniques employed, it can anyway be argued that–given the high visibility of nuraghi at landscape level–our knowledge reflects a relatively accurate picture of Nuragic landscape patterns as found in Sardinian plateaus.

A further limitation to the reconstruction of the functional dynamics of Nuragic buildings in the plateau is given by the absence of archaeozoological, archaeobotanical, pedological and pollen analyses: the closest context where a palynological analysis has been carried out is that of Pranu 'e Muru [29], where during the course of MBA and RBA significant clearance activities have been detected. However, as noted, the spatial distribution of Nuragic sites in this plateau differs significantly from that recorded in the *Giara*, thus making a direct ecological comparison inappropriate.

Such distinctive spatial properties have led to the formulation of a variety of functional and cultural interpretations. In Taramelli's [78] and in Lilliu's [80] classical accounts, nuraghi are placed as to maximize their potential for territorial control, to be intended in a basically militaristic sense. More specifically, nuraghi are thought to be placed in close proximity to the most accessible areas of the plateau's slopes, as well as in close proximity to the upper reaches of access routes on the top of the plateau. At the same time, this attitude towards territorial control is described as further enhanced by the visual control from nuraghi over these access routes. This relationship has been reiterated as significant in recent accounts [127, 128], where it has been rather interpreted as a means to grant control over *the Giara*'s resources (for example, wood and pasture), instead of as a feature of a structured "defensive" system.

Access routes to the plateau are generally referred to with the local name of "*scalas*" (stairs) [126], and have been alternatively identified with historically established, least-cost pathways leading to the top of the plateau [127] or with the dried-out beds of *the Giara*'s seasonal rivers, serving as pathways during summer [126]. The only mapping of the *scalas* found in archaeological literature is the one provided by Taramelli [78]: however, while this early recording of a still lively pastoral system is certainly relevant, and could refer to local informants, it provides no background knowledge on how *scalas* were actually identified and mapped. At the same time, the toponyms referring to the *scalas*–as found in modern cartography–correspond generally to relatively ample areas found at the edges of the plateau, rather than to specific pathways, as these routes are generally described.

A further problem regarding the mapping of the *scalas*, but more generally the reconstruction of the relationship between nuraghi at the time of their construction and the local geomorphology, is the relatively unstable nature of the Giara's edges and slopes, which are sometimes characterized by the presence of rockslides originating from the top basaltic layer of the plateau [129]. At the moment, no systematic mapping of these areas has been conducted, and a proper reconstruction of the geomorphological features of the plateau during the Bronze Age is therefore not possible. It should be noted that—to a certain extent–features such as the distance from the nuraghi to the edge of the plateau could be somewhat different than those recorded today, and should be considered for further research on the plateau.

Despite the problems in the exact definition and mapping of the *scalas*, their relevance in the archaeological context of *the Giara* has so far not been questioned. In fact, the patchy

nature of this knowledge closely matches the intuitive process by which the relationships between nuraghi and access to *the Giara* has been studied: these relationships have been described in a case-by-case fashion, rather than systematically assessed. However, in the context of this paper it is important to underline the more general archaeological significance of the *scalas*, as they are thought to represent the most accessible areas of the plateau's edges: rather than single *scalas*, what can be successfully represented and modelled is accessibility itself, the archaeological significance of which is central in current functional and cultural interpretations of the nuraghi in Sardinia's basaltic plateaus.

## Archaeological features

For its peculiar conformation and prominent archaeological features, *the Giara* has been systematically studied since the beginning of the twentieth century, starting with the pivotal mapping work of Antonio Taramelli [78].

The first excavation work was conducted in 1962 in the *archaic nuraghe* of Bruncu Madugui (Fig 2)—located in the South-Eastern corner of the plateau. While only partially published, this excavation provided a single C14 date of 3770 ± 250 uncal BP (Gif-243) [130], which, at the time of publication, proved crucial for the framing of *archaic nuraghi* as an early expression of the Nuragic phenomenon, but -being a single date and with a wide standard deviation- it is now considered as scarcely relevant. The ceramic context of Bruncu Madugui [131] has been dated to the late phases of the Middle Bronze Age. It must be stressed that data from Bruncu Madugui represent one of the very few examples of published MBA contexts, which is still widely referred to as a key context. Recent data from the *archaic nuraghe* Conca 'e Sa Cresia in the Siddi plateau have recently helped refine the absolute chronology of these buildings, placing an early phase in the 18th-early 17th century BCE [8, 27], but a consistent evaluation of their chronology is still pending, due to the scarcity of data and datings. At Bruncu Madugui, further excavations in the adjacent village [132, 133] revealed two compounds of circular huts

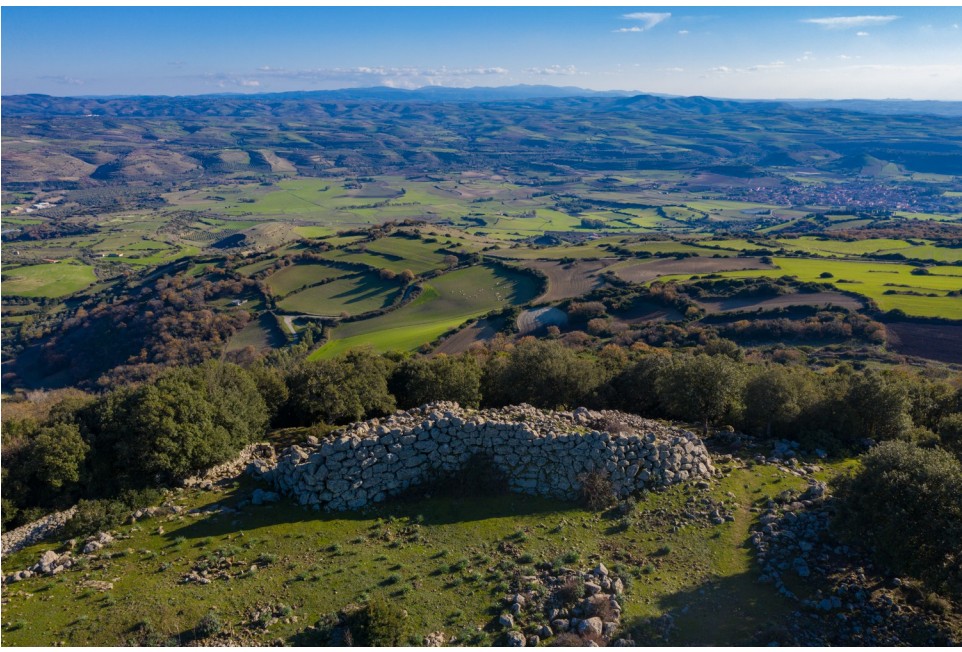

**Fig 2. View from nuraghe Bruncu Madugui (Gesturi).**

arranged around a central courtyard, as parts of a wider village. In Usai's report [133] they are interpreted as providing the first evidence of stable, year-round occupation of *the Giara*–in contrast with nuraghi, which the Author relates to a seasonal exploitation of the plateau's resources. Bruncu Madugui's hut village is in fact dated–on the basis of its ceramic context–to the Final Bronze Age (1.200–950 BCE), a period of profound change which seemingly was marked by the end of the construction of new nuraghi, and the expansion of stone-built villages, and sanctuaries. Common to both *the Giara*'s nuraghi and villages is their dependency–in terms of sustenance–from other settlements placed in nearby contexts: in fact, the basaltic soils of the plateau are not suited for large scale agriculture, while instead providing ample resources for herding activities. The high number of Bronze Age sites in the vicinities of the plateau could contribute resources [8, 127].

For this study, all sites on *the Giara* recorded in previous research have been re-surveyed, taking as a starting point the previous archaeological maps of the area [8, 78, 134], which are in turn derived from past extensive surveys of the plateau. The new survey involved the direct on-field analysis of the monuments, and their observation through aerial photographs (taken with a commercial drone or derived from Google Earth™) in order to provide an updated architectural description. A systematic survey of the area–which is planned for future work– was not carried out for this specific research. In fact, a) the overall good visibility of nuraghi ensures, to a reasonable degree, that major monuments are not missing in our record, as they would have otherwise been noticed by the numerous scholars who have worked in the area, or at least by the locals who regularly work in the area and b) comparison with previous studies is ensured, as it must be noticed that the overwhelming majority of studies on Nuragic landscapes have adopted extensive survey techniques (e.g.[135]), and the current ideas here presented derive from similar survey approaches.

A total of 20 nuraghi can be counted (Fig 3, S1 File): 16 of these display a low degree of architectural complexity (Fig 4), and are characterized by a single tower, often provided with a front-facing courtyard. Two monuments can be classified as *archaic nuraghi*: while Bruncu Madugui is clear in its charateristics, Nieddìu was classified as a simple nuraghe with a courtyard by other authors [60]. Its main structure shows instead an elliptical plan typical of *archaic nuraghi*, and it is here assumed as such, also because of its marked wall inclination, recalling Bruncu Madugui. Two nuraghi show some degree of complexity: Perdosu is characterized by two juxtaposed towers, probably connected by a short wall; Su Corrazzu (Fig 5) shows instead the presence of a central tower and a well-preserved courtyard; further lateral towers were noticed by Taramelli [78], but their exact plan remains now difficult to reconstruct–what is clear is the impression of a single case of complex nuraghe in respect to the context of *the Giara*.

Funerary structures are represented by the two Giants' Tombs of Conca 'e S'Ebba: first found during the 1980's [136], they are located at the North-Eastern end of the plateau, ca. 1.2 km southeast of Nuraghe Pranu 'e Omus.

Eleven hut villages can be counted: these are usually difficult to interpret on the field but they are characterized by localized mounds of stone rubble, among which wall remnants can be discerned. In some cases, for example in Gurdillonis [133], the arrangement of huts around a central space, typical of FBA Sardinia and such as at Bruncu Madugui village (cf. above), is visible. Hut villages are in close proximity to nuraghi in five instances, while the others remain relatively isolated (Scala Parda, Gurdillonis, Scocca Baddicchi, Cuili Frughesu, Sa Corona Arrubia, Bruncu Suergiu). Finally, the site of Sa Corona Arrubia shows the presence of a so called "reunion hut" ([137], for a discussion of this type), together with the remnants of a structure of the "rotonda" kind, likely water-related [138]: these structures, round in plan, are built with carefully shaped building blocks, and are generally related to cultic or collective

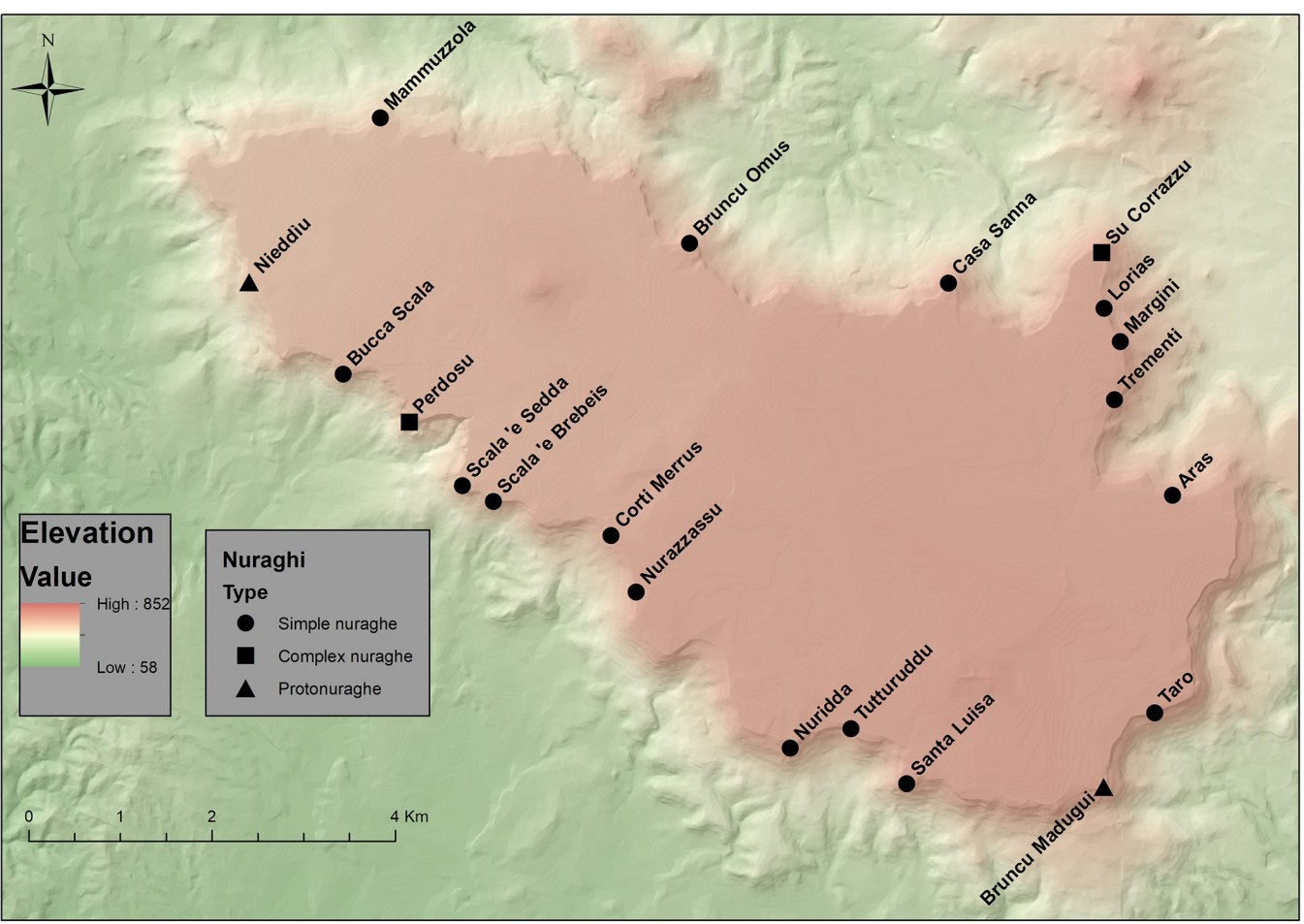

**Fig 3. Distribution and typology of nuraghi on the top of *the Giara* of Gesturi (Base map: Digital terrain model of the region of Sardinia, 10 m resolution; available at <https://webgis2.regione.sardegna.it/geonetwork/srv/ita/catalog.search#/metadata/R_SARDEG:JDCBN> under the CC BY 4.0 license; shapefile by the authors).**

"public" activities. As with hut villages, their chronology is mainly referred to the Final Bronze Age [21].

It should be stressed that archaeological data from *the Giara* of Gesturi, despite abundant in terms of monuments and villages, is still very lacking in order to provide an even preliminary diachronical reconstruction of settlement distributions and land-use patterns. In fact, as in most studies of Nuragic landscapes, chronology for nuraghi has to be assumed by expanding results from the few excavated contexts across the island, and therefore mostly from the association of the types of buildings with specific phases. This limitation however–while possibly obscuring diachronic dynamism–will have a relatively minor impact in this study, as each monument will be considered in relation to its local affordances in terms of visibility and accessibility, rather than as part of a synchronic system: in fact, measures of visibility and accessibility are taken individually for each monument, and then used for the calculation of averages. With this approach, the outcome of the analysis does not depend directly on the synchronicity of the sites, as it would for example when computing their intervisibility; at the same time, it should be noted that our averages should be taken as reflective of a long term dynamic, the chronological details of which are still to be assessed in further in-depth research on the plateau.

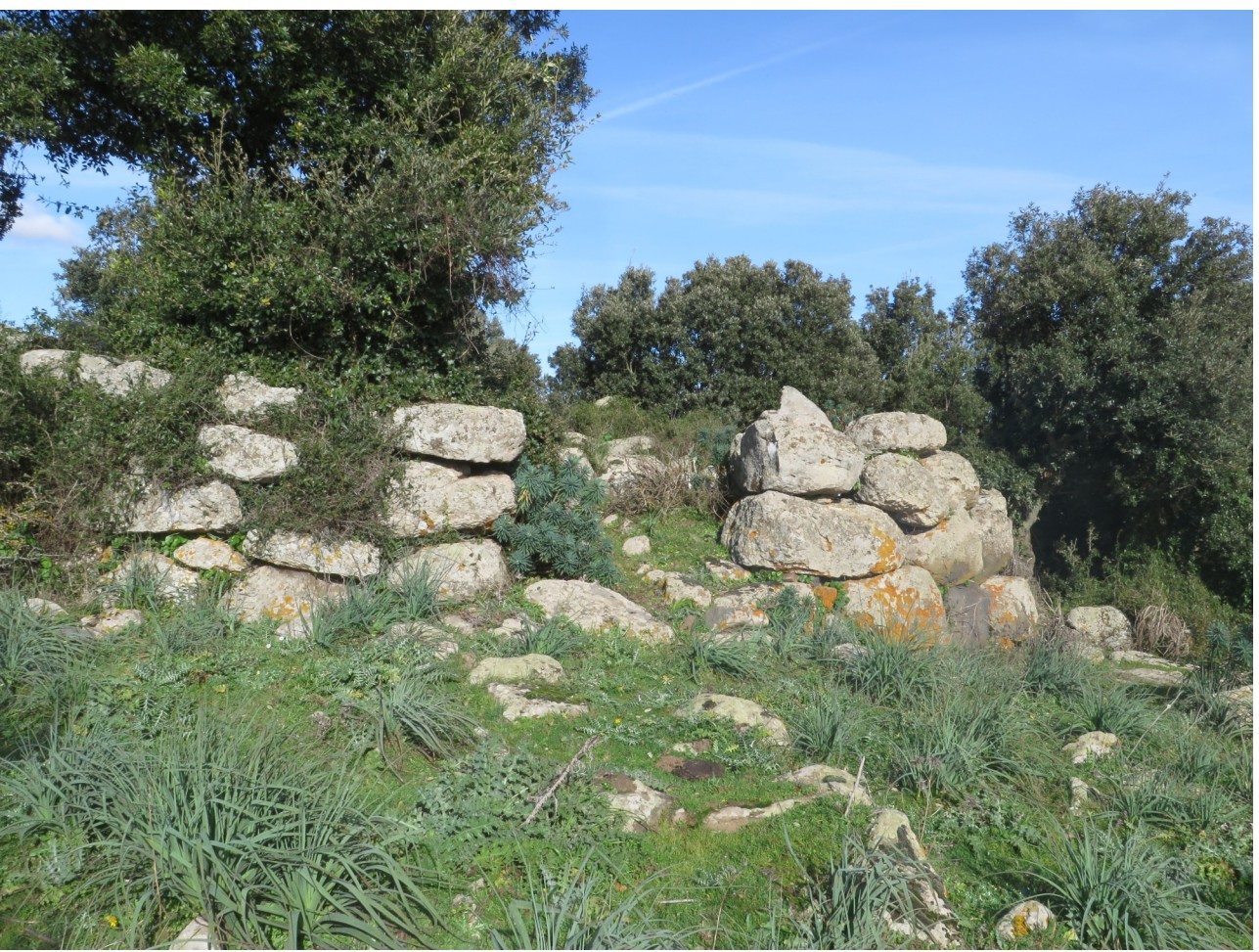

**Fig 4. Simple nuraghe Lorias (Genoni).**

## Methods

### GIS, movement and visibility

Cost Surface Analysis (CSA) and Least-Cost Path Analysis (LCPA) represent the most popular ways of studying movement in GIS-based archaeological studies (CSA: [97, 139]; LCPA: [140, 141]). CSA and LCPA have variously been used as starting points for the simulation of overall patterns of movement across landscapes or more localized movement centered around sites. The simulation of single LCPs [142], as well as of multiple one-to-many [143] or many-to-many LCPs [144–146] has been used to approximate patterns of human displacement across wide distances, while CSAs centered around sites can provide a means to assess local accessibility and defensibility [147]. The shortcomings of these techniques are well known [148]: firstly, the inclusion of cost factors such as vegetation, soil type and stream/fluvial features in calculations is very problematic, while fundamentally conditioning human movement; at the same time results can vary greatly depending on the cost function of choice [149]. This second limitation implies that studies based on CSA and LCPA should be considered explorative in nature, and caution must be exercised when evaluating results, as derived from only one of many possible models.

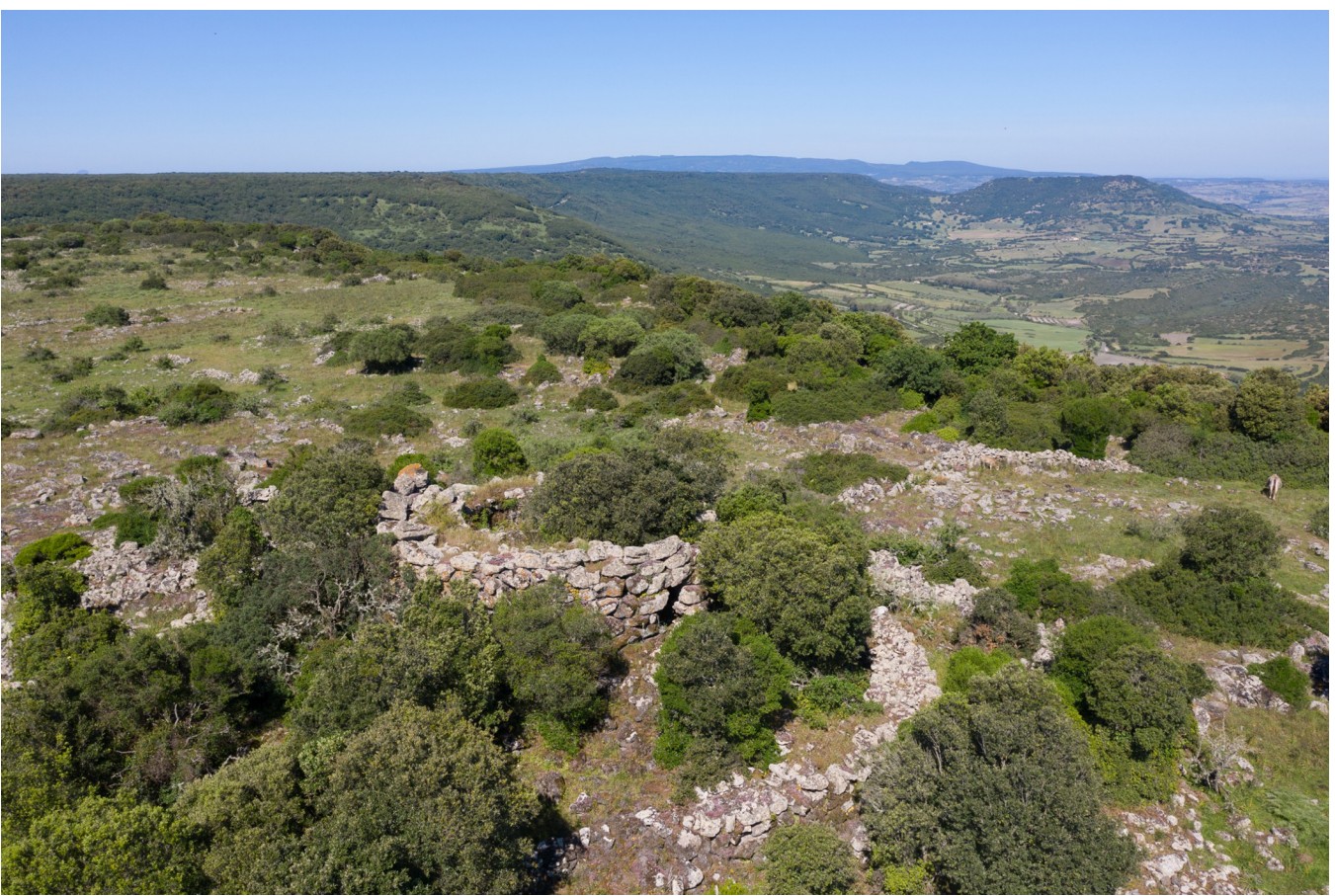

**Fig 5. Complex nuraghe Su Corrazzu (Genoni).**

The use of viewsheds as an analytical tool is today well-established, and particularly in European archaeology [150]. Specific areas of interest have been the study of defensibility and landscape control in general [151–156]. In its simplest form, a viewshed provides a binary representation of visible and non-visible areas from an observation point [97]. The most used application of this technique is the combination of multiple viewsheds in order to represent overall patterns of visibility, as in the case of cumulative and total viewsheds [111, 157]. Many of the caveats underlined for the application of CSA and LCPA apply to the use of viewsheds: while still a flexible tool, their calculation heavily relies on the accuracy of the available DTM, and it is still unclear how the problems of past vegetation and different geomorphology should be addressed, given the difficulty in modelling these factors [139, 148].

## GIS analyses

### The background mobility model

The first step of the present analysis has been the creation of a representation of potential movement directed towards *the Giara* of Gesturi. The main criterion in this model is the independence between sites and simulated movement patterns, as it is precisely their hypothetical association that has to be verified: in fact, current intuitive models of plateau access conceive low-effort established access routes (*scalas*) as pre-existing, "natural" areas of interest for

Nuragic builders [60, 78], but this remains to be tested. Therefore, in the present analysis, assumptions on where exactly movement would have started from and where it would have been directed were replaced by a general access model to *the Giara*: in order to build this model, multiple Cost Surfaces and LCPs were generated, and start and end points respectively randomly and evenly defined (see further).

The basis for the CSA and LCPA is the 10m resolution DTM provided by the Region of Sardinia [158], and slope was used as the only cost factor, as modelled in Tobler's Hiking Function [159]. Analyses were run in ArcMap by ESRI, using the Path Distance and Cost Path tools [160]. As in generalized movement models already devised (e.g. [146]), instead of specifying starting and end points based on known archaeological Nuragic sites, which would have been respectively outside *the Giara* area and on its edges, a sampling approach was adopted. This meant specifying a range of possible starting and end points, which collectively can represent a set of *all* CS/LCP possible movement patterns directed towards the plateau. First, 2500 starting points were randomly placed in a 3km wide annular buffer centered around the centroid of *the Giara*, with an internal radius of 9Km (Fig 6, S2 File). Second, a grid of evenly spaced points (100m apart) placed on the plateau was used as the destination of LCPs, for a total of 4700 end points and 11,750,000 LCPs. In other words, *each* location on *the Giara*, and not 20 nuraghi or other few sites, was considered as a potential destination–a necessary simplification, given the unknown distribution of resources or areas of interest on the plateau, apart recorded sites. This follows the assumption that access to *the Giara* was the natural goal of movers, thence relevant in terms of CS and LCP, and upon this background, site location should be evaluated. Finally, each simulated path's cell was assigned the value 1, and they were summed in order to represent their frequency across the landscape. It should be stressed that this is only one of many possible plateau accessibility models, given the multiple natural and cultural variables underpinning accessibility itself. At the same time, it limits assumptions based on natural and cultural factors to a minimum, while capturing the specific archaeological issues at hand; in turn, this is intended to minimize possible model-specific idiosyncrasies that can negatively affect the significance and the interpretation of the analyses.

The raster surface representing the (cumulative) probability of movement (henceforth MPS–Movement Probability Surface, Fig 6) was then used to extract different indexes of movement, to be further related to visibility.

## Nuraghe location and access

First, the movement frequency at the edge of the plateau (S3 File), as well as its variation as a function of the distance to the nuraghi, was considered. In this model, local high scores of MPS are held to represent the location of "natural" access routes to the top of *the Giara*: more specifically, the frequency of movement across the edge of the plateau was represented by corresponding MPS values. The correlation between nuraghi and MPS values–as recorded on *the Giara*'s edge–was carried out by checking their distance. Starting from each nuraghe, 15 groups of closest points (with a number of points ranging from a minimum of 10 to a maximum of 150, scaled by increments of 10) was selected in order to extract the corresponding MPS values: such points are located on *the Giara*'s edge and placed 10 m apart from each other, a distance chosen based on the pixel size of the DTM used (Fig 7, S4 File). For each of these point groups, MPS values associated with points were summed and averaged between the 20 nuraghi. This resulted in a total of 15 values (that is, one for each different group of points), each reflecting the average MPS distribution in smaller and larger closest point groups. The same averaged values were also calculated for 99 simulations of 20 random points located on the edge of the plateau, 20 being the number of the real nuraghi located at the edge; the

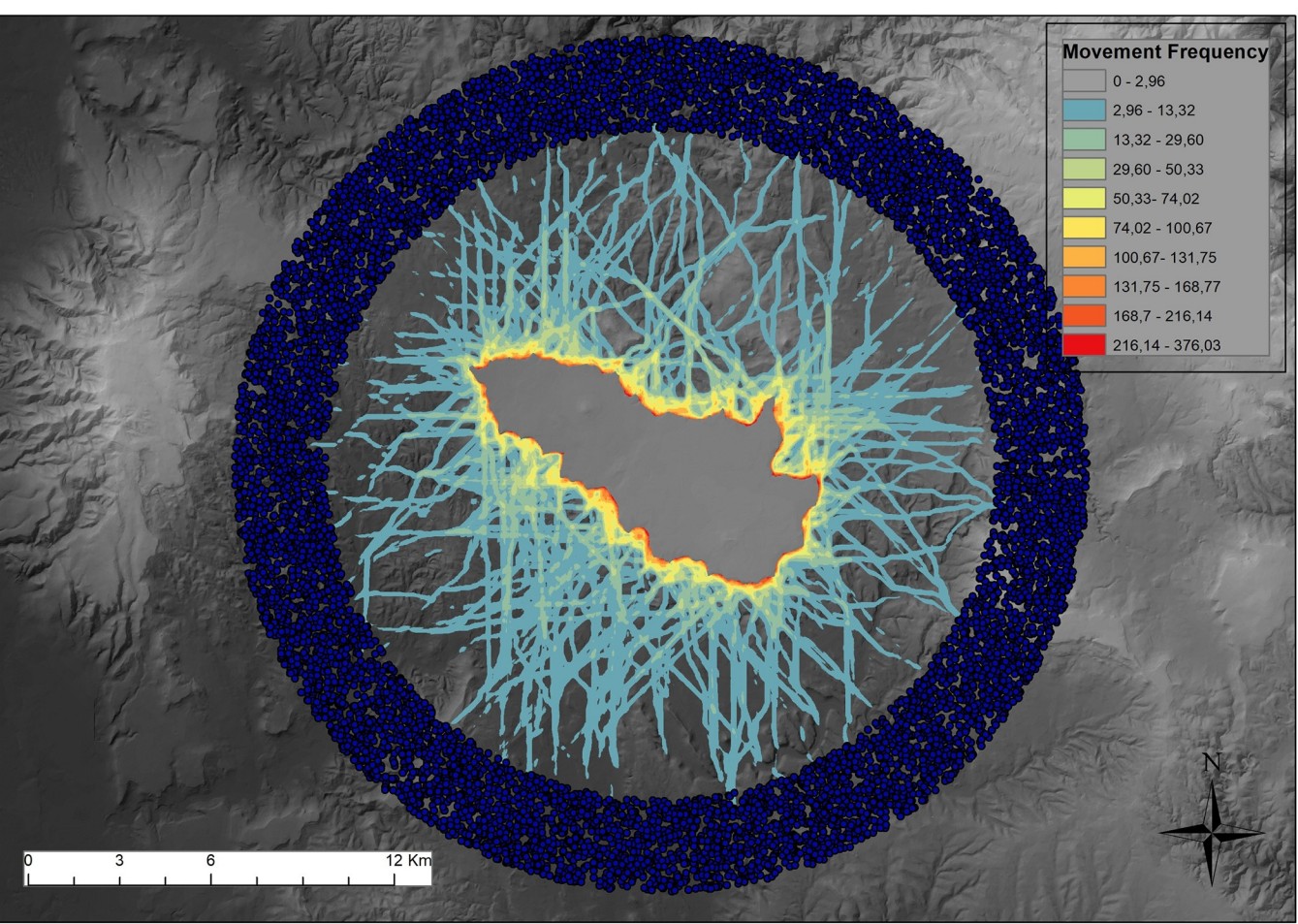

**Fig 6. Starting points for simulated paths (in blue, located in the annular buffer) and raster representing the summed values of paths (probability of movement).**

values calculated for nuraghi and for random points' simulations served as the basis for the calculation of a Monte Carlo test [161, 162].

## Nuraghe location and movement on slopes

Secondly, the correlation between nuraghi and MPS values on *the Giara*'s slopes was considered. Rather than targeting the presence of single, specific pathways leading to the plateau, this analysis considers the distribution and significance of multiple paths as distributed across areas located in proximity to nuraghi. In fact, evaluation was carried out by considering a series of 15 buffers centered around the sites, with progressive radiuses from 100m to 1500m (Fig 8, for a similar approach addressing the relationship between monuments and movement in their surroundings, see [163]), separated by increments of 100m each. For each of these buffers, the area considered is the one falling *outside* of *the Giara*'s edge, and therefore on its slopes only. The distances and the number of buffers here considered are explorative in nature: the exploration of a series of different distances can be considered as a necessary form of multiscalar analysis. The evaluation of the movement frequency for each buffer was carried out by collecting and summing the corresponding MPS values; for each 100 m distance band, the

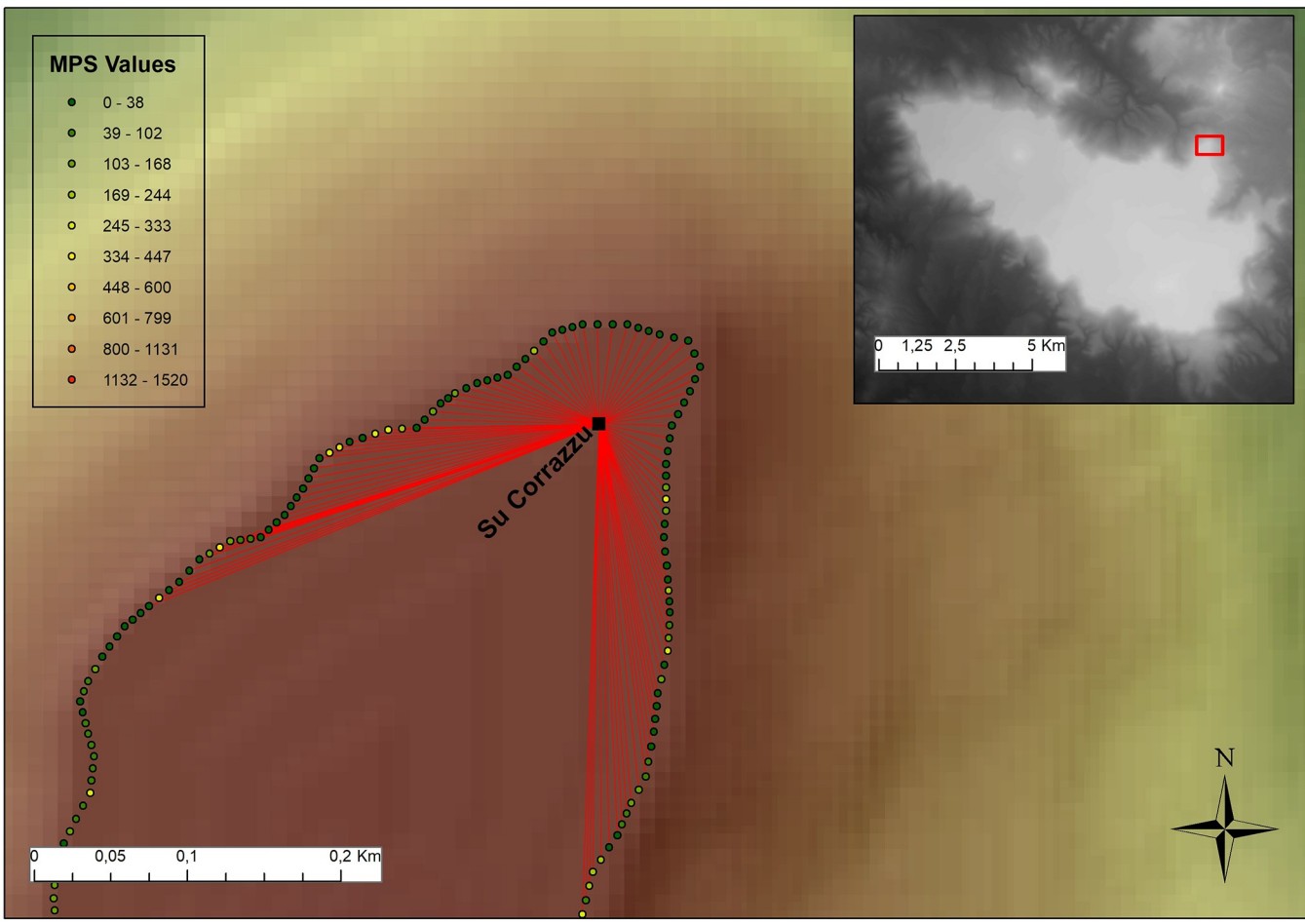

**Fig 7. Nuraghe Su Corrazzu and relative position of MPS values (example of 100 points considered) across the edge of *the Giara* plateau.** MPS values are coloured in a green-red scale.

averaged sum values correlated to nuraghi were then evaluated against the ones derived from 99 simulations of 20 random points, once again located along *the Giara*'s edge, as in the previous analysis. While the first analysis in the former paragraph is aimed to check whether the nuraghi are in connection with the precise *Giara* terminals of "natural" access routes, this second one points to their location as connected, or not, to particularly favorable access zones to *the Giara*.

It should be noted that the extension of the buffer covering the slopes of the plateau is heavily influenced by local morphology. In fact, buffers associated with sites located on outward crests are significantly wider than those associated with sites on rectilinear tracts of *the Giara*'s edge, as for crest nuraghi a smaller part of the close landscape is represented by *the Giara* itself. As a consequence, sites on crests would result in more access opportunities, given a fixed control radius. While this effect might well have been intended by Nuragic builders, it was deemed useful to normalize the previously obtained MPS sum values. This was accomplished by dividing such sum values by the effective extension of each slope buffer, thus providing an estimation of the standardized *density* of MPS scores. In other words, it is this localized *density* of movement that can be assumed as an indicator of the proximity of nuraghi to more accessible slopes.

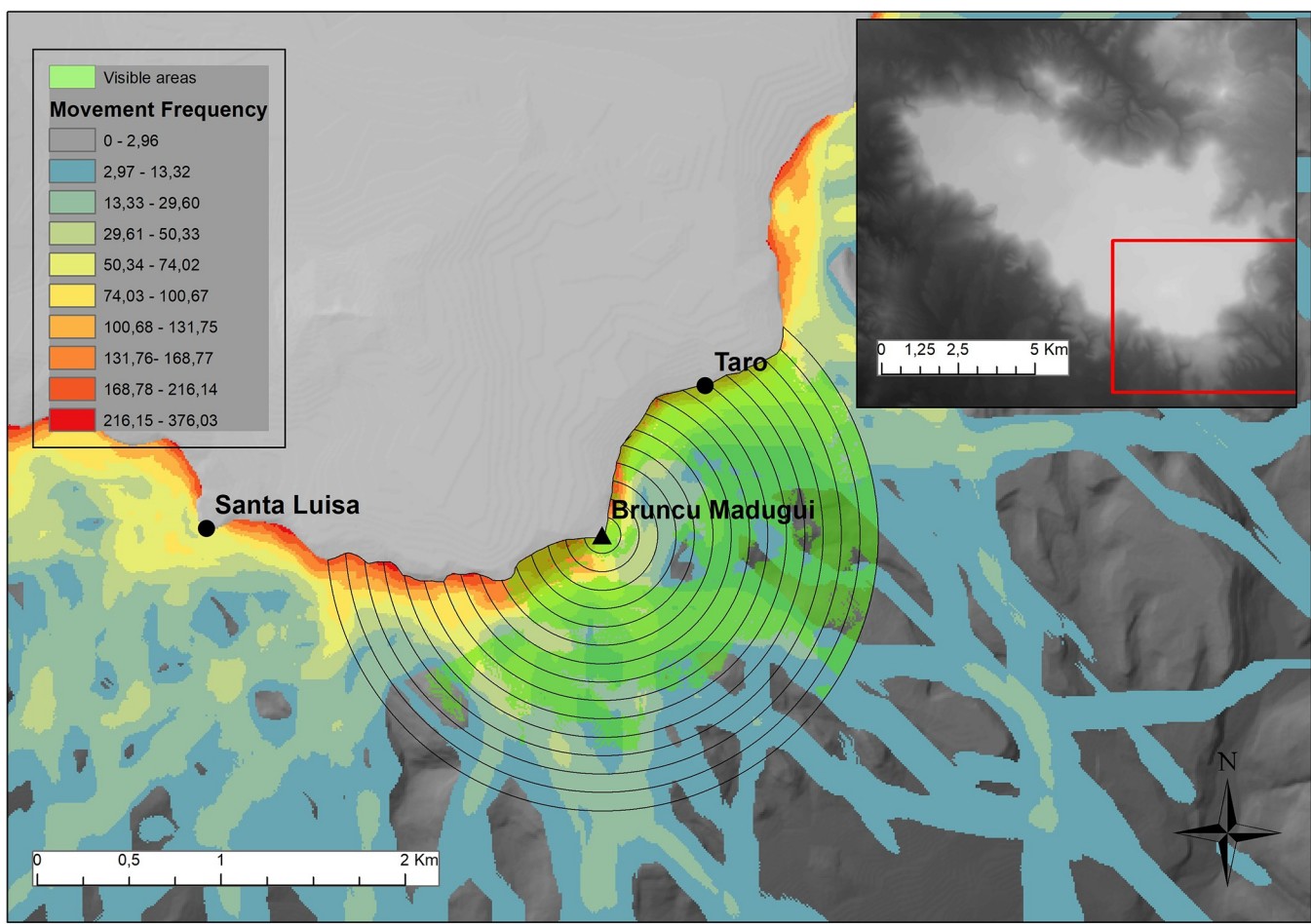

**Fig 8. Example of buffer areas considered for the analyses, relative to the MPS, with highlighted (green) visible areas.** Simple viewshed from archaic nuraghe Bruncu Madugui.

## Nuraghe location and visual control

A similar approach was then adopted in the study of visibility. First, the degree of visual control from the whole set of existing nuraghi over *the Giara*'s slopes was tested, in a way independent of MPS values. While in Nuragic studies visual control is generally assumed to be directed to the access routes [60, 78], it was here deemed useful to establish the degree by which such control could instead be a correlate of visibility *per se*, as already proposed in a former study of nuraghi at lower altitude in the same region of Marmilla [120]. Simple viewsheds were generated for each nuraghe on the plateau, again with radiuses ranging from 100m to 1500m. A standard observer vertical point offset of 10m was included in the calculation, in order to simulate the average ideal height of low-complexity nuraghi, such as those found in the plateau. In GIS-based visibility studies of nuraghi, a number of different vertical offsets has been employed [118, 120, 124, 125, 164], ranging from 10 to 20m, based on monument complexity and different summary data [165]. While the low complexity of monuments in this study area seems to warrant our conservative approach, it should again be stressed that we should think of these as rule-of-thumb measures, the real impact of which–in rather coarse GIS models–can still be assessed in future research. Once again, values of visibility were collected only for the viewshed areas located outside the plateau's edge, as the goal is to assess the

control of the slopes, and compared to a series of viewsheds generated from 99 point simulations with 20 points each. For these random points, the same 10m offset correction was applied.

Finally, the degree of visual control from nuraghi on slope "natural" access routes, as given by the MPS values, was tested; this analysis addresses more closely the established scholar assumption that the visual control granted by nuraghi was directed towards specific areas of the plateau's slopes, and namely over access routes [60, 78]. This trend in control was addressed by simply considering, for each monument, the movement buffer areas (as MPS values) visible from the corresponding viewshed (Fig 8, green areas). The averaged sum of values recorded for the 20 nuraghi was then compared with the values gathered from 99 random simulations. As in the case of simple accessibility, the values of visual control over accessibility were standardized, by dividing such values by the corresponding buffer areas.

As already underlined, each of these analyses provides a useful summary statistic, which serves as the basis of a comparison between values associated with nuraghi and random points simulations. This comparison has been evaluated by applying a Monte Carlo test [161, 162], through which the statistical significance of the difference between real and simulated values can be evaluated.

## Results

### Nuraghe location and access

The analysis of the sum values of movement frequency, as distributed on the plateau's edge, has yielded significant results: the values recorded for the nuraghi rank between the lowest and third lowest ones, when compared to the simulated ones, as for point groups ranging from 10 to 120. These low values directly reflect the presence of a lower number of paths crossing *the Giara*'s edge in the strict proximity of nuraghi, and show a minimum α significance level of 0.06 (Fig 9, S1 Fig). This result might reflect the particular morphology found in the very proximity of nuraghi, which is often rather steep, cliff-like and difficult to cross. The picture gradually changes when considering point groups from 120 to 150, which show increasingly less significant results, highlighting the local nature of the aforementioned restriction of movement, while access is instead easier at some distance from the nuraghe.

### Nuraghe location and movement on slopes

This picture seems to be confirmed when looking at the frequency of movement distributed on the plateau's slopes. When considering the smallest analytical distance, given by a buffer radius of 100m, the average value shown by nuraghi is the smallest in comparison to all the simulated values, thus seemingly confirming a tendency to local movement restriction on the plateau's slopes, as well as on the edge, close to the nuraghi. When the areas with radiuses ranging from 300m to 500 m are taken into consideration, the average values for nuraghi rank instead very high among the simulated ones (Fig 10, S2 Fig). When considering even larger areas, this tendency remains, while becoming progressively less significant. These results thus indicate the presence of a broad correlation between nuraghi and general accessibility, and potential movement as recorded on the plateau's slopes. At the same time, accessibility at shorter distances appears to be restricted in both analyses: nuraghi are therefore specifically located in areas of restricted movement, while being close to intense movement areas.

The nature of the relationship between nuraghi and movement corridors, or accessibility in general, can be further defined by looking at the standardized *density* of movement in each analysis buffer. In this case, movement still appears to be limited inside the smallest analysis radius, but no significant difference between nuraghi and simulated values can be found in wider buffers (Fig 11, S3 Fig). In other words, the significant differences found in unweighted

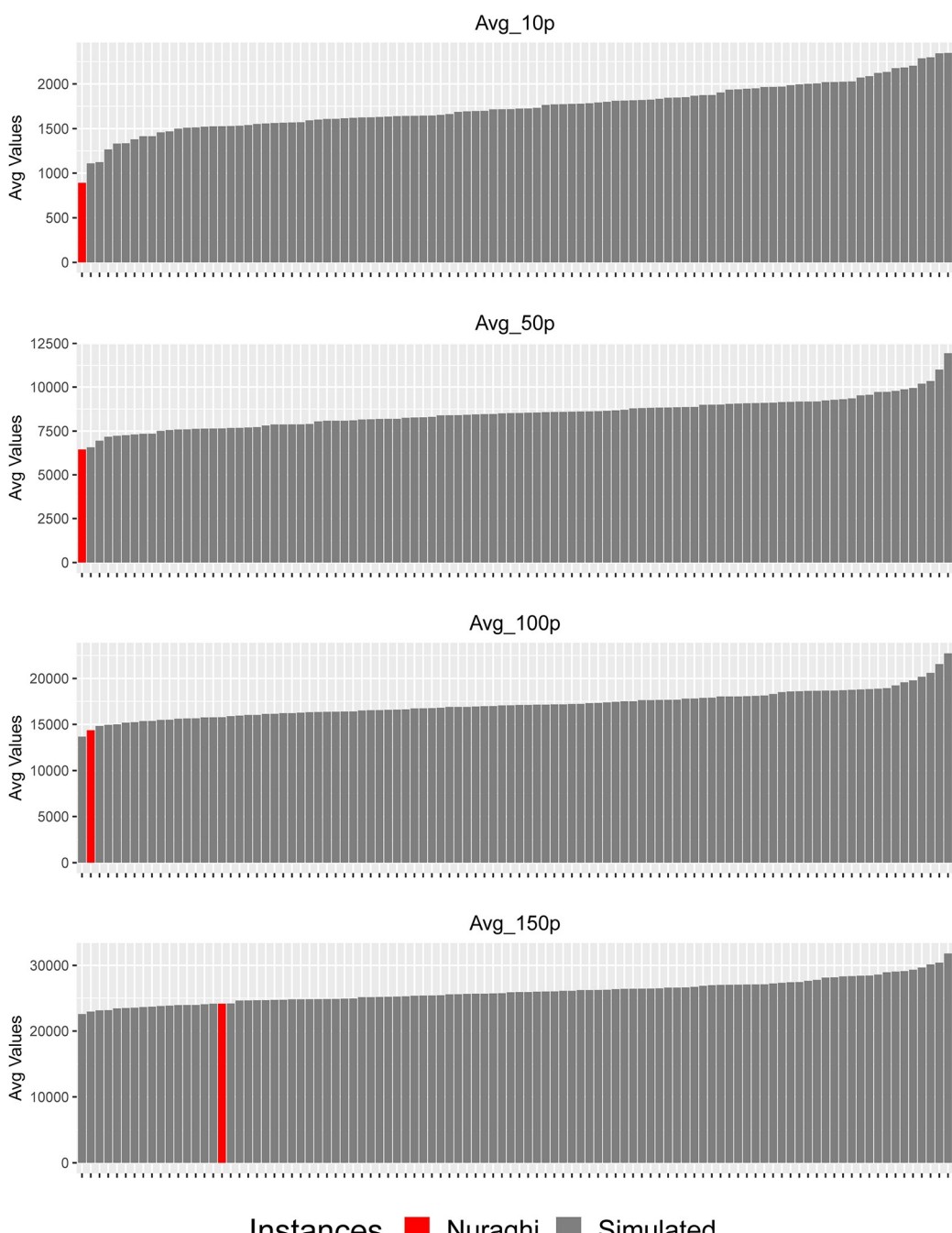

**Fig 9. Comparison of averaged summed accessibility values relative to *the Giara*'s edge, with 10, 50, 100 and 150 point groups considered.**

values seem to derive mainly from the particular crest morphology chosen for nuraghi, which granted closeness to wider parts of the plateau's edge and slopes; when considering the overall distribution of movement per area unit, this difference tends to disappear, meaning that some of the significant differences noted for unweighted value can be ascribed to the specific punctual location, rather than to differential accessibility. Therefore, secluded location was likely more important than access and movement control.

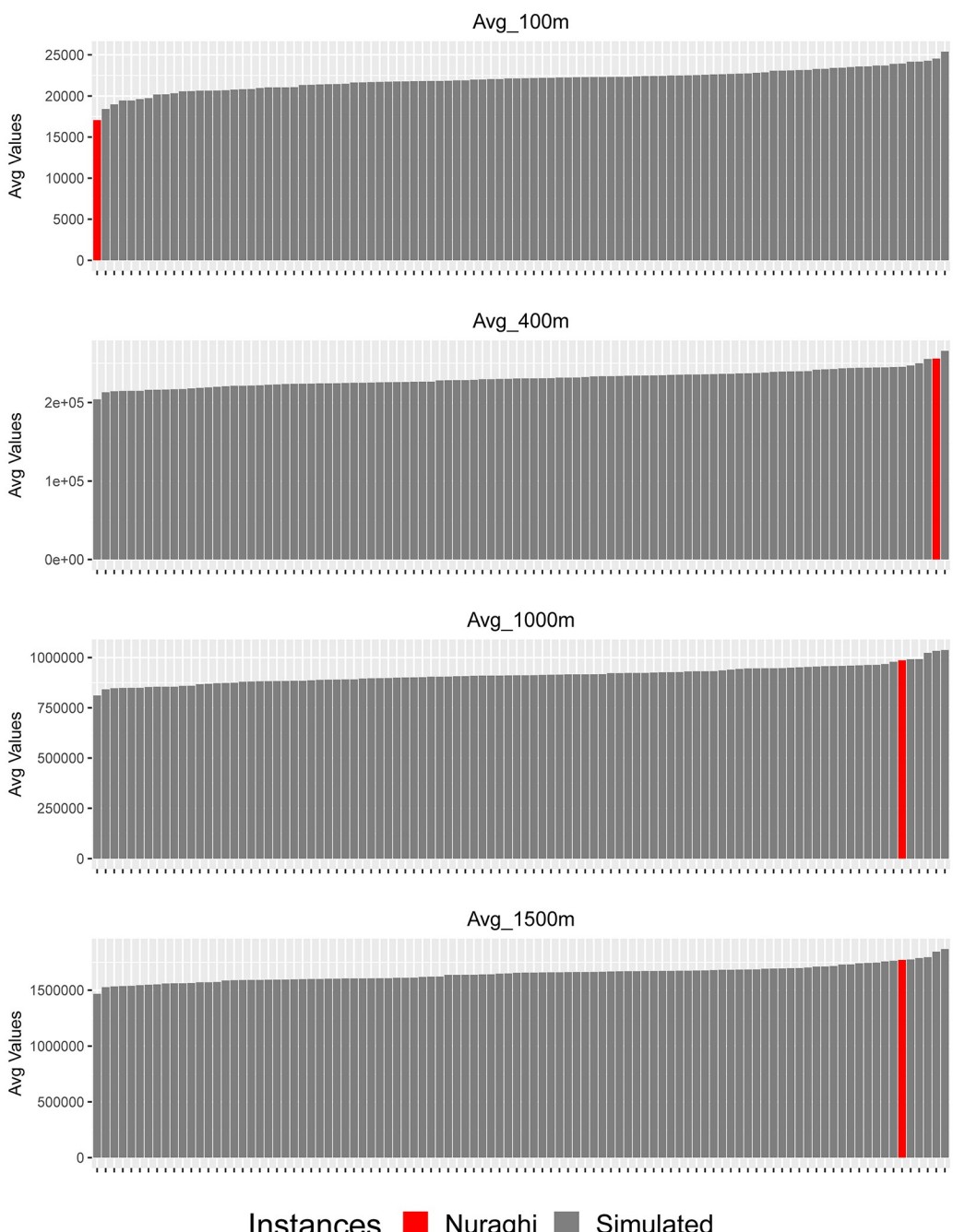

**Fig 10. Averaged movement probability values along *the Giara*'s slopes: Nuraghi and simulated point groups.** Buffer radiuses of 100 m, 400 m, 1000 m, 1500 m.

### Nuraghe location and visual control

Finally, the viewshed analysis shows significant trends as well (Fig 12, S4 Fig). When compared to random points, visibility on the plateau's slopes over the closest 100 m radius distance is scarce; such difference is significant at $\alpha = 0.04$. When considering buffers with $r \geq 400$, the average visible area is instead larger than all simulated instances. The same pattern holds true

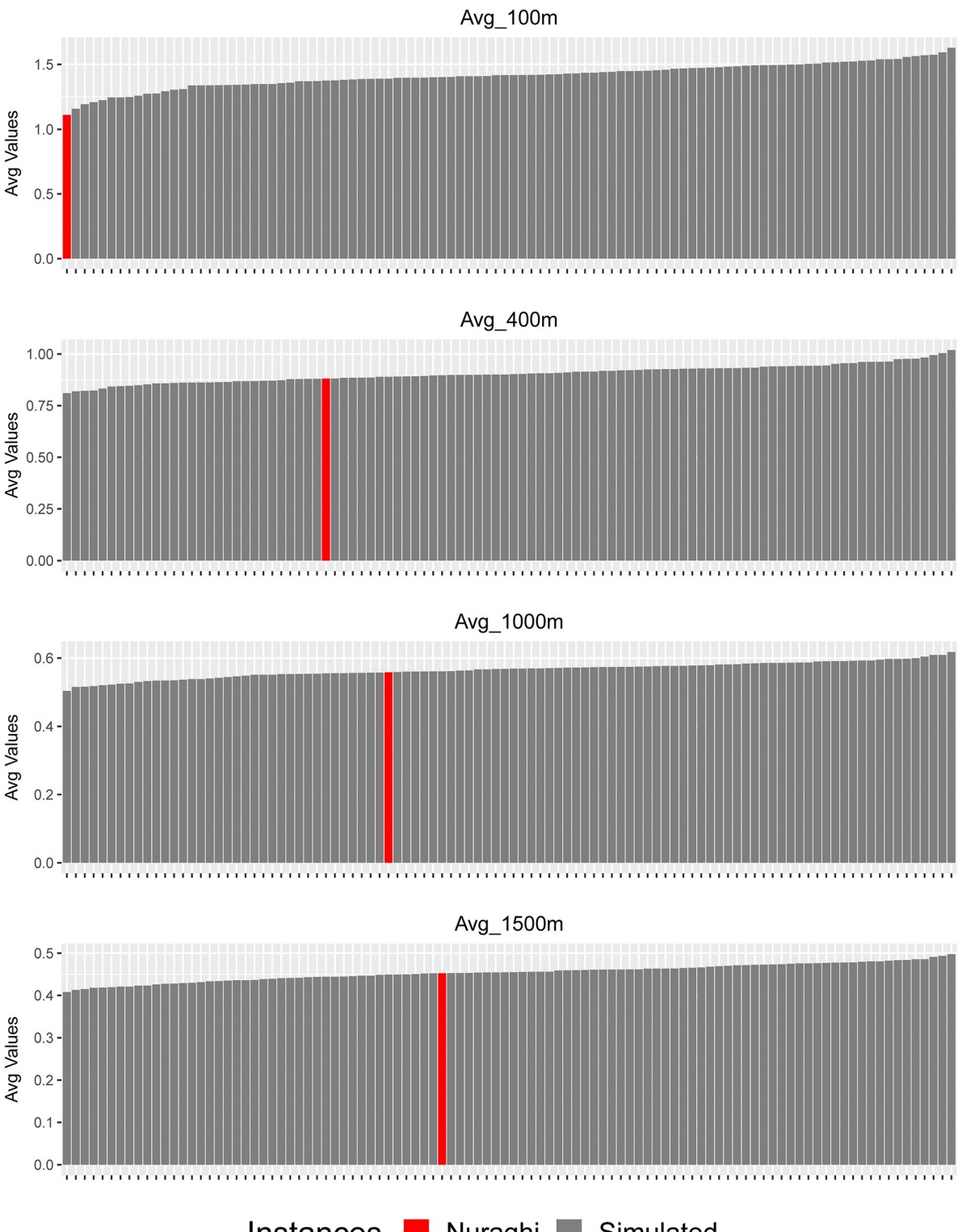

**Fig 11. Averaged density of movement probability (movement frequency divided by buffer area): Nuraghi and simulated point groups.** Buffer radiuses of 100 m, 400 m, 1000 m, 1500 m.

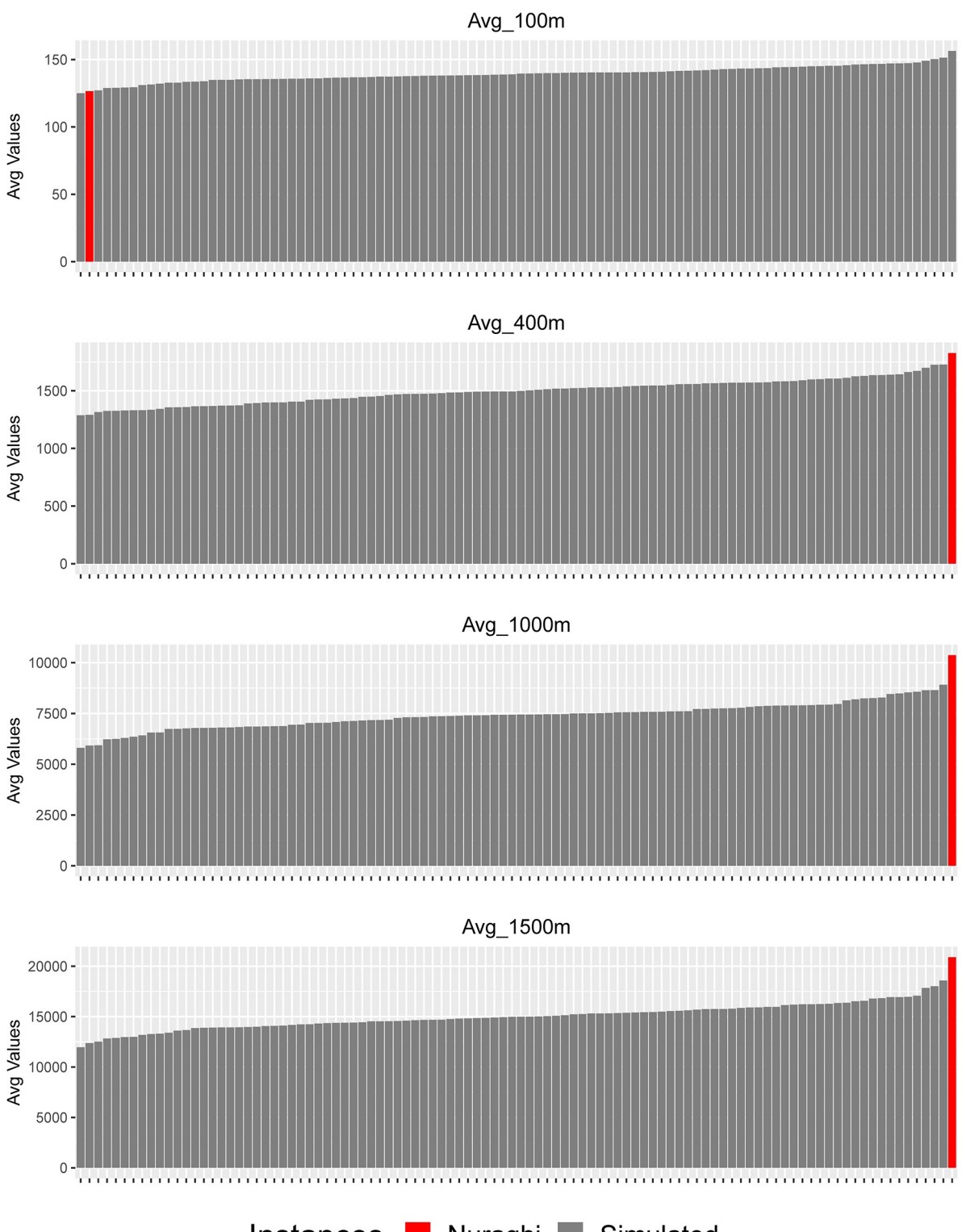

**Fig 12. Comparison of visible areas from nuraghi and random point groups, as distributed on the plateau's slopes.** Buffer radiuses of 100 m, 400 m, 1000 m, 1500 m.

when we consider the values of movement frequency, as recorded in areas visible from nuraghi (S5 Fig). It is anyway uncertain, whether this result is indeed a by-product of general visual control; that is, the main factor in this case could depend on visibility *per se* rather than on a search for visual control over specific access areas. In fact, as observed previously, the relation of nuraghi with movement frequency *per se* becomes less important as larger distances are considered, or it is blended in the general landscape control. It is useful to check if these results can, at least in part, be influenced by the local crest morphologies chosen for many nuraghi locations, as already observed in the former paragraph.

In fact–and more significantly–when considering the standardized *density* of the visual control over movement frequency, nuraghi still show the lowest recorded value for $r = 100$ (Fig 13), but results for radiuses between 300m and 500m are no more significant (see S6 Fig for complete results), thus depending on the morphology chosen for the building of the nuraghe. Conversely, nuraghi still show the highest recorded values for $r \geq 500$m. In other words, a significant visual control over localized access routes–for distances smaller than 500m- is not present or not detected by our model, and movement could instead be controlled at medium and large distances (obviously in case of a scarce vegetation cover). However, we have already seen in the former paragraph that weighted accessibility values do not appear significant at medium and large distances, as a similar distribution exists in terms of accessibility both for nuraghi and simulated points; this fact suggests to consider with prudence also the connection between visibility and movement. In fact, the correct consideration should be that it is visibility *per se* and its distribution over the landscape that can be considered the main factor affecting the values related to the standardized density of visual control, and not only over accessibility. That is, visibility in general was more crucial than visibility over movement, while this resulted as a (possibly in part intended) by-product. As a converse effect, nuraghi on the edge were potentially highly visible by people moving along the distant paths, and could act as landmarks (see below).

## Discussion and conclusions

The results of this study can be usefully contrasted with previous work on *the Giara* and the general conceptions regarding the relationships between nuraghi, movement and visibility.

In first place, the relationship between nuraghi and natural access routes, here modelled as movement frequency (MPS, cf. § GIS analyses), is more ambiguous than generally thought in scholars' assumptions. In fact, a restriction of access at shorter distances from monuments has been highlighted. While this is coherent with the trait of defensiveness traditionally linked with nuraghi [78, 80], as a sort of buffer for any threat posed against the settlement, it should not be taken as an indicator of a completely developed defensive function, but rather as an ancillary one, which could have been appreciated, anyway. At the same time, as postulated in the literature [60, 78], proximity between nuraghi and access routes has been detected by our model, but not in the sense of a strict association between a nuraghe and the route terminus. This rightfully suspected association appears to have been obtained in a spatially rather different way, and so far overlooked: in fact, the chosen locations of nuraghi–the crest positioning– afforded a better control, by proximity, over wider areas of the plateau's slopes, and thence over movement potentially directed towards the plateau, but not precisely controlling the point where routes give access to the plateau. In this sense, the builders' interest appears to be on specific plateau's morphologies, such as the outward crests, rather than on the maximization of localized control over access routes (which derives as a by-product, through potential movement control). Such conclusion is supported by the data regarding the standardized movement density, presented above. It may be stressed again that our results seem to

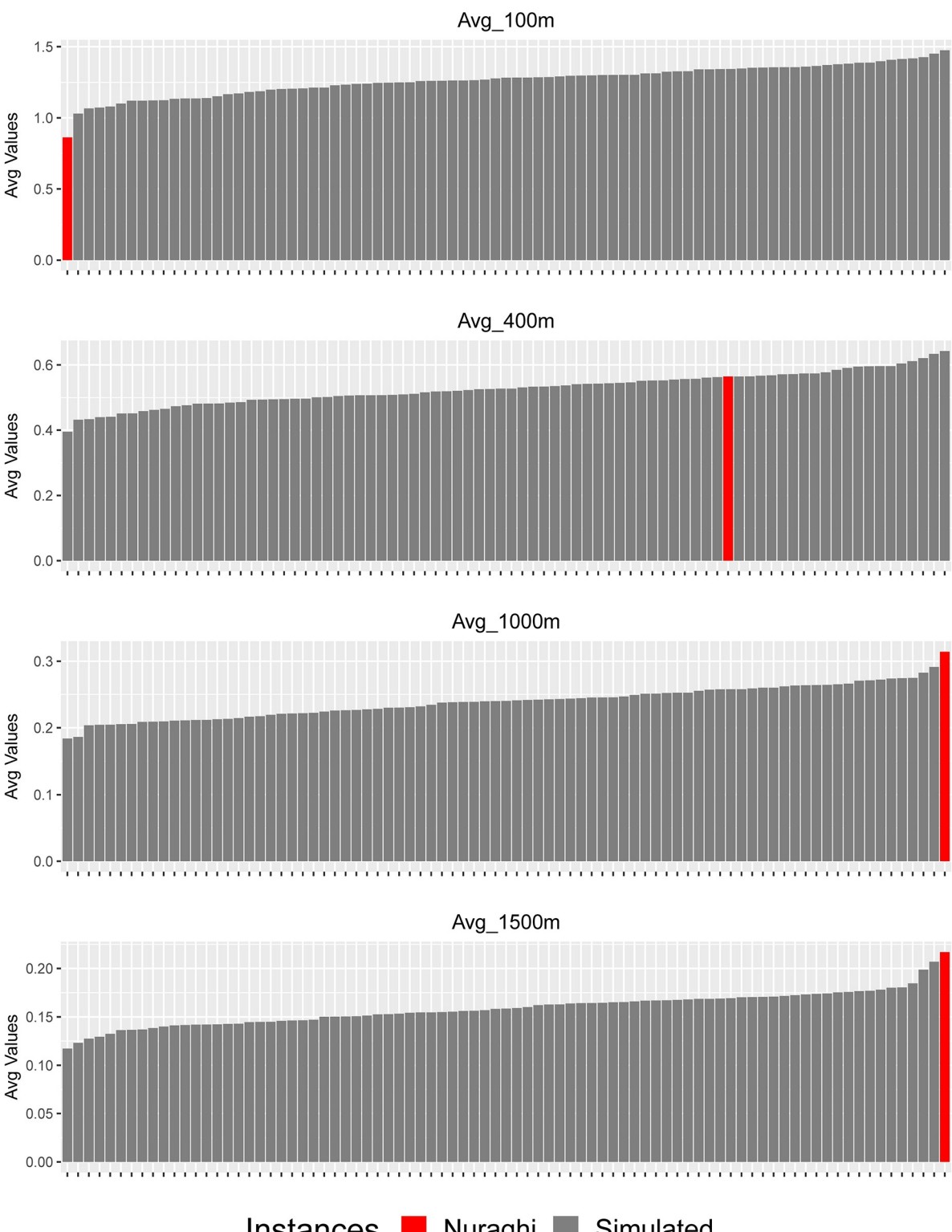

**Fig 13. Comparison of averaged values of visual control density over movement frequency (sum of values for visible paths divided by buffer area).** Buffer radiuses of 100 m, 400 m, 1000 m, 1500 m.

downplay the commonly accepted spatial proximity between nuraghi and movement corridors inside the landscape. As a future task, in order to enforce the model, another aspect could be assessed: while a relevance of control over localized access routes has not been detected by our model, a more precise definition of the characteristics of the often quoted traditional pathways -*scalas* (cf. above)- and their systematic mapping (where reliable) could possibly enrich our understanding of their spatial relationships with nuraghi. This requires direct fieldwork, as their 20th century rough mapping is not sufficient, and it can face some difficulties, due to the modern carelessness of many traditional paths.

The viewshed analysis has confirmed the crucial role of visibility in nuragic landscapes, as already established in previous studies [91, 120, 121]; at the same time, how and why this attitude towards visibility came about has been made clearer by looking at its intensity over space, and its relationship with movement (standardized density of visual control, cf. above, Nuragic location and visual control).

The focus of nuraghe locations on *the Giara*'s edge established a homogenous visual predominance over wider distances, and, as just discussed, the control over access routes appears to be, at least in part, a by-product of this tendency. In fact, when distances between 500 and 1500 m are considered, not only is the visibility from nuraghi the highest recorded in absolute, unweighted, terms, but these high values apply also to the *quantity* of movement visible per area unit. That is, while the intensity of movement alone above 500 m of distance from the nuraghe is not significantly different from the one derived from random point simulations, the weighted visibility over movement is instead higher for nuraghi in respect to the random point simulations. This sets remote visual potential as one of the most significant features of the nuraghi on *the Giara*'s edge, and three interpretations can be put forward: either visibility over remote movement is simply a by-product of location, or there was an intended control of remote movement, or what was more relevant was to be seen by remotely moving people. Combinations of these factors should be also considered.

When distance buffers within 500 m of distance from the nuraghe are considered, visibility and movement patterns are instead similar and both show significantly lower values than the random point simulations. As noted for movement *per se*, which is reduced in proximity to the nuraghe, this restriction of short-range visibility might be a consequence of the specific local morphology chosen for nuraghi (such as the outward crests, on steep slopes). More generally, it remains questionable whether the standardized values of visual control over proximal movement frequency can satisfactorily be linked to an effective territorial or even "military" control. With this regard, the comparison between absolute values of visibility and weighted values of visual control over accessibility at medium distances (300m – 500m) is instructive: while visibility *per se* is high, the degree of (weighted) visual control is not significant in comparison to simulated values, that is, the preference for visually dominant location was certainly not intended to be directed at controlling the most accessible areas of the slopes, at least at these mid- and short-distance ranges.

These results can only in a very general sense be compatible with the notion of correlation between nuraghi, movement and control thereof around *the Giara* plateau, and for the most part not in the sense envisioned in current literature [60, 127]. In fact, specific morphologies were chosen for the building of nuraghi, as well as locations affording ample visual control, as noted since the inception of Nuragic landscape archaeology (the "militarist" interpretation: [78]). At the same time, the specific nature attributed to visual control has to be questioned: its relationship with movement appears to be particularly complex, and on some occasions our results directly contradict an interest over movement control. It then follows that, while a certain tendency towards planned defense and control over territories and resources has been detected for the nuraghi of *the Giara*, their interpretation as buildings optimised for defensive

or territorial control purposes is here significantly weakened. Other social and symbolic goals could therefore account for the existing visual prominence attested on *the Giara* and more generally in Marmilla [120]: nuraghi could act as a powerful visual statement of the presence of the community, and of its right to exercise its vital activities in and around the monument itself. In light of the crowded nature of the Nuragic landscape of Marmilla, such presence would have been detectable by communities and moving people well beyond the immediate reach and neighbors of each single community, in a way that exceeded immediate concerns for safety and visual control, to be still partly detected, anyway. This is the sense in which we think it is proper to use for them the concept of "landmarks".

An open question remains, depending on the scarcity of excavation, geoarchaeological and paleoclimatic data available for *the Giara* plateau, as well as the lack of a more precise sequence of construction, use and abandonment of each single nuraghe; certainly the placement of a new tower affects the possible future choices, as nuraghi on *the Giara* rim tend to be spaced one from the other. For these reasons, a comprehensive evaluation of how spatial properties of nuraghi correlate with specific site activities and land-use patterns remains widely uncertain, and therefore speculative–thus leaving our landscape analysis preliminary, and not fully contextual. At the same time, the abundant spatial data available still provides a first informative layer on specific cultural choices, and represents a necessary part of future research on *the Giara*'s nuraghi.

The present paper has hopefully demonstrated how past and still lively traditional approaches, while providing a necessary background for current debates, and a necessary source for the generation of research questions and models, can be improved upon and integrated with the use of model-based and quantitative context-oriented analyses, formally addressing concepts such as accessibility and control, in an effort aimed at disentangling the well-known complexities of Nuragic landscapes [8]. It is in the light of this potential that the criticism levelled against formal models–by even the best scholars of empirically-based research (e.g. [61, 90])–seems at least in part unwarranted, and worth a reconsideration. In particular, the simulation-based approach here adopted, and the associated Monte Carlo tests, have proven crucial for a meaningful assessment of the spatial properties of nuraghi in *the Giara* landscape.

One could say, nuraghi are effective landmarks, whose function is not simply optimized as defensive potential, but should be seen, at least in part, as a sort of statement of location. Finally, the puzzling nature of Nuragic landscapes as reconstructed in *the Giara* plateau, should serve as a word of caution when taking intuitively assessed spatial properties at face value, beyond *the Giara* context itself and in their inclusion in broader functional and cultural narratives.

## Supporting information

**S1 File. Point shapefile of nuraghi on *the Giara*.**
(ZIP)

**S2 File. Point shapefile of randomized starting points for the LCPs directed towards *the Giara*.**
(ZIP)

**S3 File. Polygon shapefile representing *the Giara*'s edge.**
(ZIP)

**S4 File. Point shapefile representing the *quantity* of movement along *the Giara*'s edge.**
(ZIP)

**S1 Fig. Comparison of averaged summed accessibility values relative to *the Giara*'s edge, with 10 to 150 point groups considered.**
(TIF)

**S2 Fig. Averaged movement probability values along *the Giara*'s slopes: Nuraghi and simulated point groups.** Buffer radiuses of 100 m to 1500 m.
(TIF)

**S3 Fig. Averaged density of movement probability (movement frequency divided by buffer area): Nuraghi and simulated point groups.** Buffer radiuses of 100 m to 1500 m.
(TIF)

**S4 Fig. Comparison of visible areas from nuraghi and random point groups, as distributed on the plateau's slopes.** Buffer radiuses of 100 m to 1500 m.
(TIF)

**S5 Fig. Comparison of visible areas from nuraghi and random point groups, as distributed on the plateau's slopes.** Buffer radiuses of 100 m to 1500 m.
(TIF)

**S6 Fig. Comparison of averaged values of visual control density over movement frequency (sum of values for visible paths divided by buffer area).** Buffer radiuses of 100 m to 1500 m.
(TIF)

## Acknowledgments

The Authors wish to thank Andrea Di Renzoni for geostatistical discussions, Mario Schirru for assistance in fieldwork and Emily Holt, who read and discussed a preliminary version of the paper. We acknowledge the support of the Sezione di Archeologia of the Department of Ancient World Studies (Scienze dell'Antichità) of the University of Rome "La Sapienza". We further wish to thank the Academic Editor, Peter Biehl and the anonymous reviewers for their insightful comments, even the harshest ones, which we guess helped us to improve the article. All lacks of clarity or possibly questionable opinions are responsibility of the Authors.

## Author Contributions

**Conceptualization:** Davide Schirru, Alessandro Vanzetti.

**Data curation:** Davide Schirru.

**Formal analysis:** Davide Schirru.

**Methodology:** Davide Schirru.

**Supervision:** Alessandro Vanzetti.

**Writing – original draft:** Davide Schirru.

**Writing – review & editing:** Davide Schirru, Alessandro Vanzetti.

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
