## [Decision Letter · Decision Letter 0]

28 Mar 2023

PONE-D-23-01219Climbing the Giara: A quantitative reassessment of Movement and Visibility in the Nuragic Landscape of the Gesturi Plateau (South-Central Sardinia, Italy)PLOS ONE

Dear Dr. Vanzetti,

Thank you for submitting your manuscript to PLOS ONE. After careful consideration, we feel that it has merit but does not fully meet PLOS ONE’s publication criteria as it currently stands. Therefore, we invite you to submit a revised version of the manuscript that addresses the points raised during the review process. Please submit your revised manuscript by May 12 2023 11:59PM. If you will need more time than this to complete your revisions, please reply to this message or contact the journal office at plosone@plos.org. Please include the following items when submitting your revised manuscript:A rebuttal letter that responds to each point raised by the academic editor and reviewer(s). You should upload this letter as a separate file labeled 'Response to Reviewers'.A marked-up copy of your manuscript that highlights changes made to the original version. You should upload this as a separate file labeled 'Revised Manuscript with Track Changes'.An unmarked version of your revised paper without tracked changes. You should upload this as a separate file labeled 'Manuscript'.

We look forward to receiving your revised manuscript.

Kind regards,

Peter F. Biehl, PhD

Academic Editor

PLOS ONE

Journal Requirements:

2. Thank you for stating the following financial disclosure: "The paper was not funded by specific sponsors"

3. We note that Figures 2 and 3 in your submission contain map/satellite images which may be copyrighted. All PLOS content is published under the Creative Commons Attribution License (CC BY 4.0), which means that the manuscript, images, and Supporting Information files will be freely available online, and any third party is permitted to access, download, copy, distribute, and use these materials in any way, even commercially, with proper attribution. For these reasons, we cannot publish previously copyrighted maps or satellite images created using proprietary data, such as Google software (Google Maps, Street View, and Earth). For more information, see our copyright guidelines: http://journals.plos.org/plosone/s/licenses-and-copyright.

(1) You may seek permission from the original copyright holder of Figures 2 and 3 to publish the content specifically under the CC BY 4.0 license.  

**Additional Editor Comments:**

Your manuscript has now been seen by two referees, whose comments are appended below. You will see from these comments that while the referees find your work of potential interest, they have raised substantial concerns that must be addressed. In light of these comments, we cannot accept the manuscript for publication, but would be interested in considering a revised version that addresses these serious concerns.

We hope you will find the referees' comments useful as you decide how to proceed. Should presentation of further data and analysis allow you to address these criticisms, we would be happy to look at a substantially revised manuscript. However, please bear in mind that we will be reluctant to approach the referees again in the absence of major revisions.

Reviewers' comments:

Reviewer's Responses to Questions

**Comments to the Author**

1. Is the manuscript technically sound, and do the data support the conclusions?

Reviewer #1: Partly

Reviewer #2: Yes

2. Has the statistical analysis been performed appropriately and rigorously? 

Reviewer #1: Yes

Reviewer #2: Yes

3. Have the authors made all data underlying the findings in their manuscript fully available?

Reviewer #1: Yes

Reviewer #2: Yes

4. Is the manuscript presented in an intelligible fashion and written in standard English?

Reviewer #1: Yes

Reviewer #2: Yes

5. Review Comments to the Author

Reviewer #1: The authors contribute to the research on the function and context of the Sardinian nuraghi from a landscape perspective, using GIS-based methods (Cost Surface Analysis/CSA and Least-Cost Path Analysis/LCPA) and viewsheds to understand the possible role of these monuments for territorial control of the surrounding landscape. Movement through the landscape and towards the Giara of Gesturi plateau in southern Sardinia, which provides the setting for this case-study, is assessed with LCPA, CSA is used to assess accessibility and defensibility, while viewsheds address monument visibility as well as the possibility to control the surroundings. The study shows that contrary to former assumptions, the role of nuraghi in the cultural landscape of the Giara plateau (and hypothetically, also in other areas) appears to have been much more complex than simply allowing for the control of natural access and transit routes. The authors highlight the significance of the nuraghi’s visibility, especially by remotely moving people. They could not directly connect their results to advantages for a defensive/military use of the monuments (although they still assume that the latter played a role in their construction), and surprisingly, their results might in some cases even contradict any interest in movement control. Despite the naturally small spatial extent of the study, it should be an important contribution to the ongoing research of the still barely understood nuragic towers, which will help to evaluate results from upcoming landscape analyses in other regions with nuraghi on the island.

The manuscript certainly delivers a well-defined approach, the execution is properly described and the questions asked are archaeologically relevant. Nonetheless, I still have some serious issues with the manuscript. The authors appear to stress a false dichotomy between what they call “intuitive” archaeology and apparently “superior” quantitative approaches and models, instead of highlighting the importance of combined methodologies for a meaningful examination of the archaeological records. This would be especially rewarding for the study of the Sardinian nuragic period, where scientific analyses and sound landscape analyses are still in their beginnings. However, the complexity of the contexts and materials require a multi-faceted research, including the “traditional” archaeological methods.

While the original study appears to be sound (with the below mentioned major reservations), the English language is sometimes unclear and needs revision. I advise the authors work with a copyeditor to improve the flow and readability of the text. The fact that the research aims at an English speaking audience does not relativize the publications by Sardinian and other authors, especially since there are relevant papers available in English.

The figures are generally good quality, readable, and illustrate the findings of the authors appropriately. The same is true for the supplementary materials.

This supposed dichotomy affects the drawn conclusions from the nonetheless most interesting results from the case-study of nuraghi on the Giara plateau, where some obvious interpretative approaches, for example concerning the symbolic importance of the monuments, are largely ignored. Finally, this biased vision leads to partly inappropriate attacks, beyond justified criticism, against Sardinian scholars who are proficient in the field but hold divergent views on the applicability of particular modelling approaches. Instead of convincing arguments for the chosen methodology and its undisputed contribution to understanding the role and function of nuraghi as monuments that shape the Bronze Age landscape of the island, the authors use polemics (lines 299-319, 784-786) and selective quotations as well as a providing a selective bibliography. The latter has unfortunately become almost standard amongst some researchers studying nuragic Sardinia, however it has definitely to be considered bad scientific practice. The expedient discussion of relevant questions will be obstructed as long as disparate groups of scholars refuse to exchange arguments and evaluate the actual contributions of the respective “other” groups. The results and conclusions clearly show that there would be the potential to integrate the proposed landscape analysis with the knowledge and approaches that have been provided by the Sardinian scholars “under attack”. I will substantiate my criticism by summarizing each section of the manuscript:

• The Abstract is well presented and provides a good overview of the aims, results and methods of this study.

• The Introduction needs thorough revision: Firstly, an uncritical citation of Kristiansen’s “Third science Revolution” (lines 43-44) seems utterly unacceptable after the weighty criticism from paleo-geneticists and archaeologists alike, together with the expectable appropriation of Kristiansen’s assumptions by the far right. This indispensable criticism is summed up in World Archaeology, vol. 51, no. 4 (2019), especially in the editorial, in the contributions of Hakenbeck and Frieman & Hofmann, as well as by Martin Furholt (2018 & 2021). Furthermore, the aDNA data from nuragic Sardinia is far from comprehensive, and the ca. 15 or so samples from this period presented by Marcus et al. 2020 are rather to be seen as the basis of a yet to establish dataset, thus no assumptions should be based on this small glimpse on Sardinian paleo-genetics. The authors themselves admit that this “It is likely that this image (…) is somehow deceptive” (l. 46-47), so why claiming that the island “has emerged as one of the most conservative places in Europe” before there is sufficient information on this (l. 44-45)? Next, using Webster (1996) as a main reference for Nuragic Sardinia and adapting his chronology seems odd. This ignores relevant work by Sardinian scholars who know the record first hand as well as the established chronology by Lo Schiavo & Perra (2018). In the last paragraph, the scope of the research is adequately summarized.

• §2 From Monuments to Nuragic Landscapes aims at providing an overview of Nuragic archaeology but unfortunately fails to do so by using the partly outdated and heavily debated work of Gary Webster as the main source. A more differentiated view and the inclusion of relevant, also contradicting, viewpoints of scholars working in Nuragic archaeology would be necessary to complete this contextualization. The random collection on papers (l. 109, bibliography 1,2, 7-21) refers to sometimes outdated research or papers dealing with particularly narrow topics and single sites, thus leaving Webster’s works the only actual overviews. Not even one of the general works of G. Lilliu on Nuragic Sardinia is included. Much work has been done in Sicily since Leighton’s “Sicily before history” and the archaeological record of Corsica has been boosted by recent comprehensive, high-quality research. The author’s statements (lines 112-114) can hardly be acceptable by scholars working on these islands. For the contextualization and connections between them and Sardinia, there are relevant, recent publications for example by Lo Schiavo, Peche-Quilichini, Araque Gonzalez, Fundoni and Miletti, to name but a few, of which the authors do not seem to be aware of. The bibliography on Giant’s tombs as well as on the sanctuaries is deficient and partly outdated. The marginally mentioned interpretations of social relations in Nuragic Sardinia (lines 119-121), restricted to G. Webster and a paper by M. Perra that is not even centred on this issue (although he has written extensively and proficiently on the case), ignoring the contradictory but relevant ideas of Tronchetti, Russu and Araque Gonzalez, without revealing the author’s ideas on the topic, must either be perceived as listless or uninformed. However, the nuraghi are almost always discussed in their social contexts, and consequently the latter should be considered in more than merely a single line. The symbolic aspects of the nuraghi (lines 168-173) are referred to with outdated literature and do not mention the most relevant tome “Simbolo di un Simbolo” by Campus & Leonelli (2012) or new approaches by Araque Gonzalez (2021). Regarding the latter, there is no short but poignant discussion on the construction process, its possible organization and its relevance for nuragic society. The confirmed uses of nuraghi, as illustrated for example by the results from excavations at nuraghe Arrubiu at Orroli, are also being ignored. The overall omission of citations of relevant work by archaeologists who disagree with Webster’s and other British scholar’s interpretations is conspicuous. This paragraph needs thorough revision to provide a serious contextualization of the monuments.

• §3 Theoretical Issues and Landscape Approaches, sums up former landscape studies centred around the nuraghi and highlights the relevance of the author’s research. This is generally done in a comprehensive way, but it is evident that again, Sardinian scholars and their statement that rigid and abstract models are not always useful for the unique and complex nuragic landscape undergo heavy, sometimes polemic criticism (lines 208-232). The approach of the Spanish school of Granada is also criticized (lines 233-246). Finally, what is not clear to me is in how far the case study of nuraghi should be relevant for the interpretation of “Mediterranean societies” who did not build nuraghi. However, the evaluation of monument construction within prehistoric societies as a social event and collective action pointing against or towards hierarchizing processes would be relevant. This point needs clarification. The following explanation and discussion of the use of GIS (lines 256-298) is well informed and competent. However, in the second part of this section inadequate polemics and the invention of the term (Nuragic scholars’ empiricism) should be nuanced and brought forward in the form of fair, transparent criticism, including quotes and ideas and their deconstruction (lines 299-319). This is a serious problem I have with this manuscript. The promoted approach by the authors seems sound and I agree fully on its usefulness (lines 320-333).

• §4 Study area: The Giara of Gesturi is well laid out and provides all relevant information. Figures 1 and 2 are reversed and their order has to be corrected.

• §5 Methods and §6 GIS Analysis follow a clear structure and the approach (a basic GIS landscape approach) is well defined. A quote is needed in line 605 (which scholar assumes that visual control was directed towards specific areas?) and the link to bibliography online in line 500 is dead (or mistaken?). Quotes are needed for each specific criticism, otherwise it remains a blurred “opponent” that is targeted.

• §7 Results is also well structured and provides interesting insights, for example that secluded location was more important than access and movement control (lines 660-661) and that visibility over the closest 100 m radius around the nuraghi is scarce (lines 668-669), which both contradict a military use, the latter would be quite tragic in case of a siege. In the case that actually visibility per se was more important than visual control, as it is stated (lines 672-674 and 691-695), it must be considered (but is not thoroughly so in the conclusions, unfortunately) that this hints towards a strong symbolic meaning and function of the nuraghi, beyond its obvious usefulness as landmark (line 697).

• §8 Discussion and conclusions presents the intriguing results and the author’s (very cautious) interpretation. Although they did not find clear indicators of a defensive (military) use of the monuments (lines 748-750), they still do not want to exclude it, but they do not discuss the issue. I would suggest to explain why all these contradictions would still favour such an explanation of the monuments, except for the fact that Lilliu and Webster attributed a defensive function to nuraghi (lines 707-708; 763-764). Again citations are missing, for example in line 710 (“as postulated in the literature” – which?). the authors hint at a possible spatial relationship between nuraghi and the traditional pathways to the plateau – and here they seem to indicate that the denigrated “empirical2 archaeologists who have postulated this relationship were not all wrong. However, they do not mention this coincidence between both approaches. Unfortunately, they do not discuss the symbolic implications that the focus on visibility has (lines 728, 741-743). The authors interpretation of their findings unfortunately remains vague regarding their results (lines 763-768). I liked the transparency given by the authors by stating that their analysis remains preliminary and not fully contextual. They show that future work in landscape archaeology will be essential to clarify aspects of the nuraghi, however they come up with the mentioned false dichotomy again and (lines 778-782) and fail to acknowledge the potential of a combination of all knowledge. Some sentences could easily be erased from the manuscript because their only aim seems to be directly discrediting particular scholars (lines 784-786).

I recommend publication after revising these major concerns. If the authors would nuance their criticism, summarize the actual positions of the researchers they rightfully want to criticize (with proper quotes) and provide concrete arguments, the paper would improve significantly. I strongly recommend them to re-think their own position and consider if their landscape approach would not be enhanced by combining it with the results from Sardinian field archaeologists and with theoretical approaches by researchers whom they ignored for now, and joint forces might finally shed more light on the still mysterious nuraghi and their functions and meanings. A bolder statement on the interpretation of the focus on visibility by remotely moving people would also be desirable, if the authors would like to provide one.

Reviewer #2: The proposed contribution is of an excellent level, quite original, methodologically impeccable. The bibliography used is good and exhaustive. The work undoubtedly deserves publication.

However, there are some observations that can improve the contribution:

- In paragraph 2.1, as far as the Nuragic civilization is concerned, the authors speak only of the Bronze Age, while it would be appropriate to speak also of the I° Iron Age, as the authors do in other paragraphs.

- The term "classical", used in the text to define the most recent nuraghes with rooms with a "tholos" vault, should, in our opinion, be used in parentheses, so as not to give the reader who is not an expert in Nuragic civilization the doubt that they are monuments belonging to the Greek and Roman classical age.

- A paleo-geomorphological analysis of the investigated area is missing. In fact, over time, some areas may have undergone substantial changes; in fact, on the edges and slopes of the plateaus, we often witness landslides and mudslides. Even considering the importance attributed to the Scalas, i.e. the natural accesses to the plateau, a mention of this aspect would be useful, also to take it into account in the analyses.

- Equally, it would be interesting to at least mention paleoenvironmental aspects. Are there archaeozoological studies or pedological and pollen analyzes for the investigated area?

- The method of retrieving data is not well specified: the authors talk about the analysis of aerial photos, but in studies of this type the direct analysis of the monuments and possibly a systematic survey of the entire territory would also have been appropriate. Was it made? Why was it not considered appropriate to do so?

- As also mentioned by the authors, we do not have a precise chronological location of the monuments considered by the analyses, as very few sites have been the object of archaeological excavations. This is important, because the type of analysis presented would preferably require having architectures of the same phase as objects. It would therefore be necessary to better explain the reasons for the choices made and how the authors think they have solved the problem.

6. PLOS authors have the option to publish the peer review history of their article (what does this mean?). If published, this will include your full peer review and any attached files.

Reviewer #1: No

Reviewer #2: No

---

## [Author Response · Author response to Decision Letter 0]

15 Jun 2023

Response to reviewers

The Authors wish to express their appreciation of the remarks and observations by the Reviewers, which have addressed important issues, and have pointed to the necessity to be more clear and definite in the expression of their proper point of view. It is a pleasure to see that both Reviewers have accepted the GIS quantitative approach that we presented, and that each of them has recognized that the crucial parts of the analysis which is the core of the study are well organized and rather convincing (§ 4-5-6-7); the observations, mainly by Reviewer #2 about paleo-geomorphology, paleo-environment, data retrieval, and chronological detail will be addressed properly, and these observations contribute to the clarification of our arguments, we guess.

At the same point, particularly Reviewer #1, expressed some more criticism about our submitted paper, which we guess can solve, also by considering in an appropriate mode some wider literature that has been indicated by Reviewer #1. 

Criticism by Reviewer #1 is wide, and in fact, while approving the rigor of the statistical analysis and the availability of data, this Reviewer only “partly” considers the manuscript sound and that data support the conclusions.

We have therefore introduced what we deem are appropriate changes to the paper, and we present a response for all the remarks by both Reviewers. We hope this will make clear our views, while reflecting our full adhesion to the peers’ blind review principle.

Reviewer #1

General response

Before answering to each observation by the Reviewer, we think it is appropriate to make some general remarks to the provided criticism: in fact, we can identify three major arguments inside the critical observations to our submitted paper by Reviewer #1.

1. The most relevant one, in our view, regards our observation that a dichotomy seems to exist in Sardinian studies between what we call “traditional” and “quantitative” approaches to the interpretation of the landscape patterns in Nuragic archaeology. The Reviewer observes that we would be unfair in stressing these positions, and this could reach the level of polemics, and include selective referencing, or even aims at discrediting specific scholars. We would furthermore exclude the possibilities for a synthesis, and more integrated approaches, caring both the empirical/traditional views and the model-using/quantitative perspectives. We think we can clarify this issue.

With regards to the dichotomy between “traditional” and “quantitative” approaches in Sardinian landscape archaeology, we argue that it actually exists: in fact, many archaeological syntheses on Nuragic landscapes do not take into account the results of the now numerous quantitative and formal approaches to the subject. Starting from this observation, which can find evidence in the quotations and in some precise observations by some authors, particularly in years 2000s (Leonelli 2008, Perra 2008, Usai 2003, 2006), it is not the aim of this paper, nor of its Authors, to perpetrate this divide. Instead, the Authors believe that traditional approaches have really provided a vast and compelling body of knowledge, a discussion of which is unavoidable even when adopting quantitative approaches, and we state this point all over the paper, by referring since the beginning to A. Taramelli’s views, and by using the empirical syntheses, such as the works in Campus et al. 2008, Depalmas and Usai 2015 as the basis for the construction of the questions to be checked by geo-statistical tools. We therefore aim to build on traditional approaches, interpretations and ideas, rather than reject them outrightly. An outright rejection would not have resulted in the need for a translation process as the Authors have discussed (line 329). In other words, we think it is definitely right to provide an actual evaluation of the contributions derived from traditional approaches; we also think that such an evaluation we made in the submitted paper (lines 756-768) has resulted in partial confirmation and equally partial rejection: we consider this result a testimony of the strength of traditional arguments, but also of the need for their formal verification and refinement. At the same time, we think we can dilute some of the apparent criticism which could be dictated in the submitted text by the need to express clearly the positions of some scholars, in order to make clear the Authors’ constructive attitude.

Therefore, in our updated manuscript, some of the criticism has been nuanced and our appreciation for traditional work better highlighted. 

We also agree that our citations in the general parts could be seen as partial, but in our intention this had been caused by the need to give only examples of works rather than a comprehensive list. We have corrected this approach, which was not in any way meant to favor any particular interpretation, making clear what we think are different views of the situation. 

2. Reviewer #1 considers outdated our reference to Gary Webster’s views in terms of chronology and Nuragic society, and excessive our reliance on his overviews. We would also omit the views on Sardinia by other scholars disagreeing with Webster and “other British scholars”; finally, the quotation of Robert Leighton’s “Sicily Before History” (1998) would be outdated, and also the reference to Corsica would downscale the recent improvement in knowledge about the region.

We accept the criticism about the rather dated literature, but at the same time we wish to stress that we wanted to refer to the (alas, latest) handbooks and syntheses available on the prehistoric and protohistoric sequence of the areas, particularly for Sardinia and Sicily, as the context of the paper would -in our view- require a more general look for potential readers interested in the methodological framework, but not expert in the details of the areas; by the way, we consider the views by Gary Webster (or Emma Blake, or Robert Leighton, or other scholars) as certainly partial, and expression of single scholars’ views, but not out of scientific quality: when appropriate, we guessed we had introduced a balanced series of quotations for different opinions. As it seems not to be the case, we reviewed our quotations and integrated them, thus hoping to have removed this negative impression.

3. A third relevant criticism by Reviewer #1 is what would be the Authors’ interpretive position toward the symbolic and social meaning of the nuraghe as a monument. We thought indeed that a complete discussion of the social and symbolic factors implied by the construction processes of nuraghi and by the possible implications of their being a monumental preseence in the land had to be discussed in a differently targeted paper, but the criticism fosters us to be more clear and detailed. We hope to have solved this problem as well, by adding to the text some more observations on the issue, as reported afterwards.

Furthermore, as for the observation by Reviewer #1 (in the Review text and not in the answer to the bulleted question about intelligibility and standard English) that English language could be made more fluent and readable, we regret for the possible inadequacy of our style, but we can assure that the manuscript has already been proof-read by an (American-)English speaking scholar, who has now confirmed the opinion on the suitability of the text.

We now list the specific actions required by Reviewer #1, reporting and commenting our changes and response. We are particularly happy that the -sometimes harsh- criticism by this reviewer didn’t affect §6 – GIS analyses nor (almost completely) §7 – Results.

Introduction

REVIEWER’S OBSERVATION 1

Submitted text lines 43-47: The Reviewer criticizes our “uncritical citation of Kristiansen’s “Third Science Revolution””, counter-referring to some papers, namely published in World Archaeology 51/4 (2019) and elsewhere by M. Furholt (2018, 2021), which express prudence and “weighty criticism”.

The Reviewer further writes that, as the sample of ca. 15 Nuragic Sardinians presented by Marcus et al. 2020 (i.e., the study about aDNA we quote at line 46 of the submitted text) is small, it is “rather to be seen as the basis of a yet to establish dataset, thus no assumptions should be based on this small glimpse on Sardinian paleo-genetics.”

ANSWER 1

We fully understand the Reviewer’s position on aDNA studies, and we agree with the prudence that is called for, in front of the small number of samples. As for the papers expressing reserve about the use of aDNA data (and analyses), we know them and particularly appreciate the thorough scrutiny that Martin Furholt gave of the Yamnaya migration hypothesis. We furthermore would also recommend to take care of the tools employed for tracing aDNA data, such as the Admixture approach (see, e.g. Lawson et al. (2018), NATURE COMMUNICATIONS (2018) 9:3258).

In any case, there are two papers on aDNA, so far, regularly published in scientific journals, which independently stress the same conclusion, i.e. that human remains found in Sardinia -after Early Neolithic times- show a major contribution from external aDNA only after Nuragic/Bronze Age times, that is, there is an intense genetic and demographic continuity in the island. They are:

- the quoted Marcus et al. 2020, with 16 samples of Nuragic age, and comprehensively 48 well-dated individuals from Neolithic to Nuragic age;

- the not quoted (but now quoted: see below) Fernandes et al. (2020), NATURE ECOLOGY & EVOLUTION 4: 334-345, adding ca. 15 samples dated to the Bronze age and comprehensively 30 from Neolithic to Bronze age (with some limits in radiocarbon dating, etc.).

Therefore, the long-stated even modern DNA peculiarity of Sardinian inhabitants (e.g., Francalacci et al. (2013), Science 341 (6145): 565-569; Sikora et al. (2014), PLoS Genet 10(5): e1004353), seems to have a relation with the island’s relative isolation, which is affirmed by different samplings of aDNA, and traced until the Iron Age/Phoenician contact. Also recent syntheses, albeit expressing doubts about the supposed Early Neolithic input of Anatolian/Aegean aDNA, repeat the same considerations (Calò et al. (2021), Annals of Human Biology 48/3: 203-212).

ACTIONS 1

Summing up:

a) we deem that a quotation of the present scientific agreement about a significant genetic/population/demographic continuity fron Pre-Nuragic to Nuragic times is appropriate for our paper’s introduction, and we maintain it;

b) We downscale a bit the quotation of Kristiansen’s “Third Science Revolution”, by postponing it in the text, and by raising some righteous care;

c) As our paper is not a DNA (neither ancient nor modern) paper, a thorough discussion such as in Calò et al. (2021) is out of scope and would deflect the paper from its main goal. Therefore, we only quote Marcus et al. (2020) and we add Fernandes et al. (2020);

d) We still claim, as observed by Reviewer #1 that “It is likely that this image of Sardinia as a separated enclave is somehow deceptive …”.

REVIEWER’S OBSERVATION 2

Submitted text lines 51-53: the Reviewer states that we would ignore “relevant work by Sardinian scholars who know the record first hand as well as the established chronology by Lo Schiavo & Perra (2018).”

ANSWER AND ACTIONS 2

We do agree that the general reference to only Webster 1996 is outdated; we have explained above (General response #2) why we had decided to introduce this reference. We maintain it anyway, but we add 2 references (6-7 in text) to chronologies established by Sardinian and other scholars (Lo Schiavo & Perra 2018, suggested also by the Reviewer, as well as Vanzetti et al. 2013, where Nuragic chronological problems are discussed by relevant authors).

Chapter 2

GENERAL OBSERVATION §2: the Reviewer remarks that, in this paragraph, the “overview of Nuragic archaeology [...] unfortunately fails to do so by using the partly outdated and heavily debated work of Gary Webster as the main source”.

ANSWER AND ACTIONS, GENERAL §2: We have explained above (General response #2) why we decided to use Webster as a formal reference (both Webster 1996 and Webster 2015). Indeed, it is our opinion that §2 does not refer only, nor mainly, to Webster’s views, but that it depends on our own direct experience in the study of Nuragic Sardinia, as well as much on a paper that we quote, but don’t want to overstress, also to avoid self-referncing, namely Vanzetti et al. 2013. 

As the Reviewer has perceived this feeling, of dependence on a single (relevant) scholar, we introduced more quotations (such as in NEW MANUSCRIPT lines 96-99) and expressed in many points of the text the different positions.

§ 2.1

REVIEWER’S OBSERVATION 3

Submitted text’s lines 97-101; 118-119

The Reviewer considers the bibliography on Giants’ tombs and sanctuaries “deficient and partly outdated”.

ANSWER AND ACTIONS 3

The bibliography on Giants’ tombs and sanctuaries has been integrated with further and up-to-date references. 

REVIEWER’S OBSERVATION 4

Submitted text’s lines 109-110

The Reviewer considers the references presented here as a “random collection on papers (l. 109, bibliography 1,2, 7-21) refer[ring] to sometimes outdated research or papers dealing with particularly narrow topics and single sites, thus leaving Webster’s works the only actual overviews”.

ANSWER AND ACTIONS 4

While we do agree that the idea of somehow “random collection” is real, as it was intention of the Authors to give an idea of the wide range of papers and views that Nuragic impressive archaeology has fostered, we don’t want to give any impression that Webster’s works are the only existing overviews. For this reason, we widened the series of quoted papers, and we inserted some basic Sardinian literature, such as, e.g. Lilliu’s founding syntheses.

REVIEWER’S OBSERVATION 5

Submitted text’s line 114

The Reviewer remarks that “Much work has been done in Sicily since Leighton’s “Sicily before history” and the archaeological record of Corsica has been boosted by recent comprehensive, high-quality research.”

ANSWER AND ACTIONS 5

The Autors agree with the Reviewer, but argue that, in the frame of the submitted paper, a further discussion of Sicily and Corsica would be out of scope, and therefore referencing had been selective and synthetic (as expressed above, General response #2). Anyway, the two cases are different:

As for Sicily, the mass of work published after the syntheses by Sebastiano Tusa (1983 and 1999) and Robert Leighton has not resulted in any synthetic overview, and therefore one has to refer still to these publications. We added the quotation of Tusa’s synthesis, as Tusa 1999 and of a recent Conference on the Prehistory of Sicily. Sicily, as stated also by our research in Cannatello, is characterized by a notable internal variability of cultural manifestations and monumental landscapes, both at a synchronic and diachronic dimension.

As for Corsica, the new works have provided some synthetic overviews, of which we are aware, but it is clearly fieldwork by Peche Quilichini (and somehow Matteo Milletti) that has provided the best new results. In any case, Corsica monumentality of tower-like buildings is not widespread all over the island, and is characterized by a considerable variability. As the reference to Corsica has also the function, in the paper’s frame, to trace the divergent trajectories of Sardinia and Corsica- in other terms why we can discuss of Sardinia (and of a specific region inside it), without necessarily debating Corse monumentality - we consider the best option to add a single paper by Peche Quilichini, precisely discussing this issue (Peche Quilichini 2021). We don’t quote here Vanzetti et al. 2013, which had evaluated the situation of Corsica and other major islands of the Mediterranean, as we consider it already slightly outdated.

REVIEWER’S OBSERVATION 6

The Reviewer argues: “The marginally mentioned interpretations of social relations in Nuragic Sardinia (lines 119-121), restricted to G. Webster and a paper by M. Perra that is not even centred on this issue (although he has written extensively and proficiently on the case), ignoring the contradictory but relevant ideas of Tronchetti, Russu and Araque Gonzalez, without revealing the author’s ideas on the topic, must either be perceived as listless or uninformed. However, the nuraghi are almost always discussed in their social contexts, and consequently the latter should be considered in more than merely a single line.”

ANSWER AND ACTIONS 6

We do agree with the Reviewer that the discussion of Nuragic social relations was compressed, but at the same time we remark that this derived from our precise knowledge of the debate, as this debate is so varied that even a whole paper could not include all Authors’ opinions. We still think that hierarchic and heterarchic are the main poles (and the best examples) between the scholars’ view of Nuragic society dwindle, but we added the anarchic and cooperative CONTROLLI LEI SE VA BENE view that has been recently expressed by Araque Gonzalez (presently quotations RIPORTARE NUMERI ora 46, 47, 74). As we agree that an ampler discussion is more appropriate, we have further expanded, as requested, the discussion on different social interpretations of Nuragic Sardinia. We have anyway avoided to insert a deeper discussion on FBA-EIA social dynamics, since such discussion would lie somewhat outside the scope of the paper, which is more centered on MBA-RBA landscapes. 

§ 2.2

REVIEWER’S OBSERVATION 7

This observation by the Reviewer is linked to what expressed above, as the argument discussed at General response 3, encompassing also other points in the submitted paper. 

“The symbolic aspects of the nuraghi (lines 168-173) are referred to with outdated literature and do not mention the most relevant tome “Simbolo di un Simbolo” by Campus & Leonelli (2012) or new approaches by Araque Gonzalez (2021). Regarding the latter, there is no short but poignant discussion on the construction process, its possible organization and its relevance for nuragic society. The confirmed uses of nuraghi, as illustrated for example by the results from excavations at nuraghe Arrubiu at Orroli, are also being ignored.”

ANSWER AND ACTIONS 7

We accept the observation, and we give now a more comprehensive overview of symbolic interpretations of nuraghi, updating the references and basically including the Reviewer’s proposals.

As noted above (Answer and actions 6) for the social reconstruction, we avoided a deeper review of symbolic meanings of nuraghi in FBA-EIA (and beyond), when scholars generally underline a symbolic shift in many archaeological contexts inside the nuraghi, because such discussion would lie somewhat outside the scope of the paper, which is more centered on MBA-RBA landscapes.

Chapter 3

§ 3.1

REVIEWER’S OBSERVATION 8

The Reviewer identifies, in our critique, a polemical vein, stating the following: “Sardinian scholars and their statement that rigid and abstract models are not always useful for the unique and complex nuragic landscape undergo heavy, sometimes polemic criticism (lines 208-232). The approach of the Spanish school of Granada is also criticized (lines 233-246).”

ANSWER AND ACTIONS 8

In the first part of the paragraph, we have abandoned some of the observations that Reviewer #1 identified as polemical in our critique: while the main points of our observations remain intact, the merits of the traditional approaches have also been better underlined. 

Our critique of the Spanish school of Granada has been slightly nuanced, by rephrasing some sentences. At the same time, some of its shortcomings had already been underscored by the same Spanish authors, whose critique we had also quoted. 

REVIEWER’S OBSERVATION 9

The Reviewer states: “what is not clear to me is in how far the case study of nuraghi should be relevant for the interpretation of “Mediterranean societies” who did not build nuraghi.”

ANSWER AND ACTIONS 9

The Authors do not believe that the analysis of Nuragic societies is immediately informative of any other processes happening in other Mediterranean contexts; at the same time, the high richness of monumental and spatial data about the Nuragic world seems to have the potential to become and represent a rather complete comparative model, even if chronological details have some degree of uncertainty. It is in this sense that we consider it relevant for the study of Mediterranean societies in general. We hope to have made this point clearer in the new manuscript. 

§ 3.2

REVIEWER’S OBSERVATION 10

The Reviewer, similarly to the previous observation, states that “in the second part of this section inadequate polemics and the invention of the term (Nuragic scholars’ empiricism) should be nuanced and brought forward in the form of fair, transparent criticism, including quotes and ideas and their deconstruction (lines 299-319).”

ANSWER AND ACTIONS 10

This section has been extensively re-written, in order to better specify the Authors’ point of view and to provide a more detailed and nuanced presentation of our criticism towards traditional landscape approaches. First of all, relevant examples of work towards which our criticism is addressed are provided, with a careful description (in a necessarily synthetic way); secondly, rather than focusing on some of what we do perceive as limitations of traditional approaches, we have stressed how current ideas and formal approaches could actually be integrated. This was the original idea of the paper, and we think now we made sure that our criticism did not obscure this original intent. 

Chapter 4

REVIEWER’S OBSERVATION 11

“Figures 1 and 2 are reversed and their order has to be corrected.”

ACTION 11

The order of figures 1 and 2 has been reversed in the uploaded documents. 

Chapter 5

REVIEWER’S OBSERVATION 12

“A quote is needed in line 605 (which scholar assumes that visual control was directed towards specific areas?) and the link to bibliography online in line 500 is dead (or mistaken?). Quotes are needed for each specific criticism, otherwise it remains a blurred “opponent” that is targeted.”

ACTION 12

Quotes have been added specifically where requested and more generally in cases where they better needed reiteration. The dead link has been substituted with a working one. 

Chapter 7

REVIEWER’S OBSERVATION 13

The Reviewer suggests that a more in-depth discussion of the possible symbolic meaning of nuraghi should be put forward, by stating that “In the case that actually visibility per se was more important than visual control, as it is stated (lines 672-674 and 691-695), it must be considered (but is not thoroughly so in the conclusions, unfortunately) that this hints towards a strong symbolic meaning and function of the nuraghi, beyond its obvious usefulness as landmark (line 697).”

ANSWER AND ACTION 13

We have chosen to further discuss the possible symbolic role of nuraghi in the following Discussions section of the paper (Chapter 8). This new discussion partly corresponds to the General response #3 discussed above.

Chapter 8

REVIEWER’S OBSERVATION 14

The Reviewer observes that our position towards the militaristic view of nuraghi is not entirely clear and justified by the data, by stating that “Although they did not find clear indicators of a defensive (military) use of the monuments (lines 748-750), they still do not want to exclude it, but they do not discuss the issue. I would suggest to explain why all these contradictions would still favour such an explanation of the monuments, except for the fact that Lilliu and Webster attributed a defensive function to nuraghi (lines 707-708; 763-764).”

ANSWER AND ACTIONS 14

With regards to our interpretation of the nuraghi in a militarist sense, we have made our position clearer. We think that to a certain degree nuraghi show spatial features that could -broadly speaking- be used for site protection purposes (visual control at medium and large distances, as well as their secluded position, notwithstanding their tower-like appearance). However, as already noted, the visual control lacks the specificity that one would expect and that has been hypothesized by the Authors who favor these interpretations. In fact, both the classic “militarist” interpretation and the one seeing nuraghi as means of “territorial control” are significiantly weakened in the light of our results, but some degree of “defensiveness” appears to be actually enhanced, especially with regards of the secluded position of the monuments. This detected degree of defensivess (joining distant visual control and local defensiveness thanks to the seclusion) is still not enough, in our view, to make nuraghi monuments optimised for militar defense or even territorial control. Instead, as already stated, other goals should be considered; conversely, one should consider also the possibility to be seen from faraway: in this respect, we have broadened our discussion of the possible symbolic value of the visual aspects of Nuragic towers, while also rephrasing and expanding our general point of view.

REVIEWER’S OBSERVATION 15

“Again citations are missing, for example in line 710 (“as postulated in the literature” – which?).”

ACTION 15

Quotes for “as postulated in literature” have been added. 

REVIEWER’S OBSERVATION 16

According to the reviewer, we “seem to indicate that the denigrated “empirical” archaeologists who have postulated this relationship [i.e., between nuraghi and the traditional pathways to the plateau] were not all wrong.” and thence we would fail to “mention this coincidence between both approaches”, that is our and previous approaches.

ANSWER AND ACTIONS 16

Our text stated “At the same time, as postulated in the literature, proximity between nuraghi and access routes has been detected by our model, but not in the sense of a strict association between a nuraghe and the route terminus”. The coincidence was therefore mentioned, with the caveat that such association could have been obtained in a spatially different way. We have made slight modifications to our wording and sentences, but we still feel that the overall meaning and intention remain intact, with an important difference between the results of the two different approaches.

REVIEWER’S OBSERVATION 17

“If the authors would nuance their criticism, summarize the actual positions of the researchers they rightfully want to criticize (with proper quotes) and provide concrete arguments, the paper would improve significantly. I strongly recommend them to re-think their own position and consider if their landscape approach would not be enhanced by combining it with the results from Sardinian field archaeologists and with theoretical approaches by researchers whom they ignored for now, and joint forces might finally shed more light on the still mysterious nuraghi and their functions and meanings.”

ANSWER AND ACTIONS 17

In the second-last paragraph, rather than remarking some of the shortfalls of traditional approaches, as the reviewer highlights and criticizes, we have opted to better stress the contribution of these empirically-based scholars, both in general and specifically for the construction of the arguments discussed (as well as for the results suggested) by our paper, where traditional interpretations have served as a necessary basis for further testing. A critical reappraisal of valid and solid hypotheses is the core goal of our contribution; we feel that this was in part (and admittedly, implicitly) demonstrated by the translation process (line 382 of the new manuscript) from traditional ideas in spatial and quantitative models. 

REVIEWER’S OBSERVATION 18

“Some sentences could easily be erased from the manuscript because their only aim seems to be directly discrediting particular scholars (lines 784-786).”

ANSWER AND ACTIONS 18

While firmly rejecting such an accusation, we hope that it originated from an unfortunate wording. The sentence has been changed accordingly. 

Reviewer #2

§ 2.1

REVIEWER OBSERVATION 1

In paragraph 2.1, as far as the Nuragic civilization is concerned, the authors speak only of the Bronze Age, while it would be appropriate to speak also of the I° Iron Age, as the authors do in other paragraphs.

ACTION 1

References to Early Iron Age have been added where appropriate.

§ 2.2

REVIEWER OBSERVATION 2

The term "classical", used in the text to define the most recent nuraghes with rooms with a "tholos" vault, should, in our opinion, be used in parentheses, so as not to give the reader who is not an expert in Nuragic civilization the doubt that they are monuments belonging to the Greek and Roman classical age.

ACTION 2

The term “classical” has been substituted with “typical”.

§ 4.1

REVIEWER OBSERVATION 3

A paleo-geomorphological analysis of the investigated area is missing. In fact, over time, some areas may have undergone substantial changes; in fact, on the edges and slopes of the plateaus, we often witness landslides and mudslides. Even considering the importance attributed to the Scalas, i.e. the natural accesses to the plateau, a mention of this aspect would be useful, also to take it into account in the analyses.

ANSWER AND ACTIONS 3

Further observations on the partly unstable geomorphological nature of the Giara plateau have been added. However, there is no specific study allowing for a paleogeomorphological reconstuction, which should in turn be taken in consideration by future research. 

REVIEWER OBSERVATION 4

Equally, it would be interesting to at least mention paleoenvironmental aspects. Are there archaeozoological studies or pedological and pollen analyzes for the investigated area?

ANSWER AND ACTIONS 4

There are no specific archaeozoological, palynological and pedological studies for the Giara of Gesturi. The closest available data comes from the Pranu ‘e Muru plateau, but a direct comparison between the two contexts could be misleading. These observations have been added to the manuscript.

§ 4.2

REVIEWER OBSERVATION 5

The method of retrieving data is not well specified: the authors talk about the analysis of aerial photos, but in studies of this type the direct analysis of the monuments and possibly a systematic survey of the entire territory would also have been appropriate. Was it made? Why was it not considered appropriate to do so?

ANSWER AND ACTIONS 5

A more complete description of the data retrieval methods has been provided, together with a better justification of the survey methods here adopted, as an effective ground check of all the nuraghi of the area has been performed by one of the Authors (D.S.). 

REVIEWER OBSERVATION 6

As also mentioned by the authors, we do not have a precise chronological location of the monuments considered by the analyses, as very few sites have been the object of archaeological excavations. This is important, because the type of analysis presented would preferably require having architectures of the same phase as objects. It would therefore be necessary to better explain the reasons for the choices made and how the authors think they have solved the problem.

ANSWER AND ACTIONS 6

As requested, we have expanded our discussion on the issue of chronology: we do agree that a better chronological resolution would be preferable, but we think that the analysis still remains useful even in its absence (further details in the manuscript). The only caveat is that our results should be taken as the average of long term dynamics, the details of which are still to be assessed.

---

## [Decision Letter · Decision Letter 1]

10 Jul 2023

Climbing the Giara: A quantitative reassessment of movement and visibility in the Nuragic landscape of the Gesturi plateau (South-Central Sardinia, Italy)

PONE-D-23-01219R1

Dear Dr. Vanzetti,

We’re pleased to inform you that your manuscript has been judged scientifically suitable for publication and will be formally accepted for publication once it meets all outstanding technical requirements.

Kind regards,

Peter F. Biehl, PhD

Academic Editor

PLOS ONE

Additional Editor Comments (optional):

Reviewers' comments:

Reviewer's Responses to Questions

**Comments to the Author**

1. If the authors have adequately addressed your comments raised in a previous round of review and you feel that this manuscript is now acceptable for publication, you may indicate that here to bypass the “Comments to the Author” section, enter your conflict of interest statement in the “Confidential to Editor” section, and submit your "Accept" recommendation.

Reviewer #1: All comments have been addressed

Reviewer #2: All comments have been addressed

2. Is the manuscript technically sound, and do the data support the conclusions?

Reviewer #1: Yes

Reviewer #2: Yes

3. Has the statistical analysis been performed appropriately and rigorously? 

Reviewer #1: Yes

Reviewer #2: Yes

4. Have the authors made all data underlying the findings in their manuscript fully available?

Reviewer #1: Yes

Reviewer #2: Yes

5. Is the manuscript presented in an intelligible fashion and written in standard English?

Reviewer #1: Yes

Reviewer #2: Yes

6. Review Comments to the Author

Reviewer #1: I am very pleased to see that the authors addressed all comments, provided sound answers to each criticism and used the rationale behind them to significantly enhance this paper.

My admittedly harsh criticism of what I considered polemic statements in the first version has been adressed and is now minimalized by mostly clearer wording and better contextualization. The term "nuragic scholar's empiricism" is in my opinion still a bit difficult and lines 299-319 seem still slightly polemic, however this might (hopefully) spark a fruitful discussion with the adressed scholars.

I really appreciate the extended interesting discussion on the scalas (lines 369-386) as wel as the more profound discussion of results. The bibliography and English language have both been significantly impoved. Personally, i would not let stand Kristiansen on aDNA uncommented or without citing his critics, but this is the author's decision.

I look forward to seeing this important and pioneering contribution to nuragic archaeology in print soon.

Reviewer #2: The authors responded satisfactorily to the suggestions and questions. The requested improvements have been made in the text. For me the text is now publishable.

7. PLOS authors have the option to publish the peer review history of their article (what does this mean?). If published, this will include your full peer review and any attached files.

Reviewer #1: No

Reviewer #2: No

---

## [Editor Report · Acceptance letter]

26 Jul 2023

PONE-D-23-01219R1 

Climbing *the Giara*: a quantitative reassessment of movement and visibility in the Nuragic landscape of the Gesturi plateau (South-Central Sardinia, Italy) 

Dear Dr. Vanzetti:

I'm pleased to inform you that your manuscript has been deemed suitable for publication in PLOS ONE. Congratulations! Your manuscript is now with our production department. 

Kind regards, 

on behalf of

Dr. Peter F. Biehl 

Academic Editor

PLOS ONE